# Environmental regulation of toxin production in *Bacillus anthracis*

**Ankur Bothra**[1]*, **Andrei Pomerantsev**[1], **Benjamin Schwarz**[2], **Anupam Mondal**[3], **Eric Bohrnsen**[2], **Nitika Sangwan**[1], **Kaitlin A. Stromberg**[2], **Mahtab Moayeri**[1], **Qian Ma**[1], **Rasem Fattah**[1], **Sundar Ganesan**[4], **Catharine M. Bosio**[2], **Stephen H. Leppla**[1]

1 Laboratory of Parasitic Diseases, National Institute of Allergy and Infectious Diseases, National Institutes of Health, Bethesda, Maryland, United States of America, 2 Laboratory of Bacteriology, National Institute of Allergy and Infectious Diseases, National Institutes of Health, Hamilton, Montana, United States of America, 3 Neurobiology, Neurodegeneration, and Repair Laboratory, National Eye Institute, National Institutes of Health, Bethesda, Maryland, United States of America, 4 Biological Imaging Section, Research Technologies Branch, National Institutes of Allergy and Infectious Diseases, National Institutes of Health, Bethesda, Maryland, United States of America

* ankur.bothra@nih.gov

## Abstract

Pathogenic *Bacillus anthracis* strains carry two plasmids - pX01, which encodes a tripartite protein exotoxin complex (PA, LF, and EF); and pX02, which encodes a poly-D-gamma-glutamic acid capsule. A multidomain transcription factor, AtxA, regulates the expression of these virulence genes. AtxA has two DNA-binding Helix-Turn-Helix (HTH) domains, two phosphoenolpyruvate: carbohydrate phosphotransferase system regulatory domains (PRD1 and PRD2), and a putative EIIB domain (a component of PTS sugar transport EII-complexes). Previous studies showed that glucose and $CO_2$ increase AtxA-dependent toxin gene transcription, along with histidine phosphorylation of PRD1 and PRD2. Our transcriptional profiling of virulence factors, PA secretion, and fluorescent reporter strain analyses confirms a synergistic effect of glucose and $CO_2$ on AtxA-dependent toxin production. Deletion of AtxA (ΔatxA) significantly reduced glucose uptake in bacteria, suggesting that AtxA may act within the glucose-PTS system. Mutation analysis of the EIIB domain of AtxA identified the cysteine at position 402 as essential for the transcriptional activity of AtxA. Deletion of glucose PTS permease PtsG (ΔptsG) significantly reduced the expression of PA, LF, and EF. Loss of PtsG also caused attenuation in a mouse model of infection. Intracellular imaging using FLIM confirms a physical interaction of PtsG and AtxA through EIIB domain of AtxA. Using phosphomimetic and phosphoablative mutants of AtxA, we confirmed that the physical interaction of PtsG and AtxA is essential for AtxA activity. Finally, the synergy between glucose and $CO_2$ was targeted by deleting pyruvate carboxylase Pyc (Δpyc), which regulates anaplerosis. This deletion confirms that Pyc stimulates the level of phosphoenolpyruvate (PEP) and increases the phosphorelay in glucose-PTS to enhance AtxA activity. Therefore, we propose that a

**Data availability statement:** All relevant data generated or analyzed during this study are included in this published article [and its supplementary information files].

**Funding:** This research was supported by the Intramural Research Program of the National Institutes of Health (AB to SHL). The contributions of the NIH author(s) are considered Works of the United States Government. The findings and conclusions presented in this paper are those of the author(s) and do not necessarily reflect the views of the NIH or the U.S. Department of Health and Human Services. The funder had no role in study design, data collection and analysis, decision to publish, or preparation of the manuscript.

**Competing interests:** The authors have declared that no competing interests exist.

histidine-phosphorelay from PEP regulates AtxA *via* PTS enzymatic activity, impacting AtxA activity through physical interaction of AtxA and PtsG. Finally, we propose AtxA as an integral component of the glucose-PTS, where transcriptional activity of AtxA is regulated by environmental signals including glucose and $CO_2$.

---

## Author summary

*Bacillus anthracis*, the bacterium that causes anthrax, produces potent toxins that are central to its virulence. How environmental signals like glucose and carbon dioxide ($CO_2$) control toxin production has remained unclear. Here, we reveal a key mechanism by which the master virulence regulator, AtxA, integrates environmental and metabolic cues to control toxin gene expression. Using transcriptomics, metabolomics, and protein-protein interaction analyses, we show that AtxA directly interacts with the glucose-specific phosphotransferase system (PTS), specifically with the glucose permease PtsG. This interaction, mediated through AtxA's EIIB domain, is essential for its transcriptional activity and toxin production. Furthermore, pyruvate carboxylase (Pyc) enhances phosphoenolpyruvate (PEP) availability, strengthening PTS-dependent signaling and boosting AtxA activation under glucose and $CO_2$ conditions. Disruption of *ptsG* significantly reduced toxin expression and bacterial virulence in mice. Together, our findings uncover how *B. anthracis* senses and responds to host-like environments to regulate toxin production, revealing an important link between metabolism and virulence.

## Introduction

Anthrax is a zoonotic disease caused by *Bacillus anthracis*, which can infect hosts through inhalational, ingestion, or cutaneous infection. The bacteria persist as desiccated spores in soil and germinate upon entering a host, developing into encapsulated vegetative cells which disseminate via the host's systemic system into various tissues [1]. To establish infection, *B. anthracis* produces anthrax exotoxins: protective antigen (PA), lethal factor (LF), and edema factor (EF). These toxins are crucial for the bacteria's virulence and ability to cause disease. *B. anthracis* utilizes signaling pathways to detect and respond to environmental changes such as temperature, nutrient availability, and aeration when transitioning from soil to host tissues [2]. The regulation of virulence factor expression is essential for the bacteria's adaptation and survival in different environments.

The most severe form of anthrax infection is inhalation anthrax. Once the bacteria breach the host epithelial barrier, anthrax toxins and other major virulence factors (poly-D-gamma-glutamic acid capsule and secretory proteases) play critical roles in impairing the local immune response [3,4]. The *B. anthracis* pXO1 plasmid encodes the tripartite protein exotoxin complex (PA, LF, and EF) and a multidomain transcription factor, AtxA, which positively regulates the expression of several structural genes and anthrax toxin [5,6].

Previous studies highlighted the importance of glucose and carbon dioxide in anthrax toxin production [7–9] suggested an indirect association between the carbon catabolite repression protein CcpA and glucose availability on the expression of *atxA* and the toxin genes. Several studies [7,9–12] have attempted to identify how *B. anthracis* utilizes glucose- and carbon dioxide-sensitive pathways to mediate signals and activate anthrax toxin production. However, none of these studies could demonstrate the role of specific genes or pathways in directly regulating *atxA* and the toxin genes.

The importance of sugar and carbon dioxide is not unique to *B. anthracis*. The dependency of bacterial virulence on carbohydrate metabolism has been long debated. Several zoonotic and human pathogens regulate expression of their virulence genes based on the type of carbohydrate they sense in their environment [9,13,14]. In Firmicutes like *B. anthracis*, the phosphoenolpyruvate (PEP) phosphotransferase system (PTS) couples the import and phosphorylation of preferred (glucose) and alternative carbon sources through sugar-specific membrane channels. A series of PTS proteins (EI, Hpr, and sugar-specific EIIA) are involved in this coupled reaction [15]. The activity of PTS proteins and CcpA is dependent on the relative abundance of ATP, early glycolytic intermediates like fructose-1,6-bisphosphate, and PEP [16]. The abundance of these metabolites has also been shown to modulate the phosphorylation of PTS-regulatory domain (PRD)-containing pathogen virulence regulators, called PCVRs [13,17–20].

Based on secondary domain analysis and the presence of conserved histidine residues H199 and H379, AtxA is classified as a PCVR, with its transcriptional activity regulated by phosphorylation of these histidine residues [13,21]. AtxA has two DNA-binding HTH domains, two PTS regulatory domains (PRD1 and PRD2), and a putative EIIB domain (a component of PTS sugar transport EII-complexes) [21]. We and others have shown that phosphorylation of H199 of PRD1 is necessary for AtxA activation while phosphorylation of H379 in PRD2 is inhibitory to toxin gene transcription [21,22]. However, the cellular systems responsible for AtxA phosphorylation and dephosphorylation are not known. Recently, a study by Bier *et al.* 2020 aimed to understand the role of PTS enzymes on regulating AtxA phosphorylation and its transcriptional activity [7]. However the work failed to clearly define the mechanism by which central carbon metabolism regulates AtxA and toxin production. Therefore, in this report, we provide data to support that synergy between glucose metabolism and anaplerosis is essential for anthrax toxin production and that a physical interaction of AtxA with glucose specific PTS (PTS$^{glu}$) proteins is pre-requisite for attaining its transcriptional activation property.

## Results

### Anthrax toxin production is dependent on central carbon metabolism

In this study, transcriptomic analysis (RNAseq) identified a total of 291 genes differentially expressed between bacteria grown aerobically in nutrient rich NBY-G medium (NBY + 10 mM glucose) or in the NBY-G medium with 0.9% (w/v) NaHCO$_3$ and grown in air supplemented with 5% CO$_2$ ("air" vs "CO$_2$") (S1 File for list of differentially expressed genes). Transcriptional changes between the air and CO$_2$-grown bacteria are compared in Fig 1A. A total of 120 upregulated (red) and 171 downregulated (blue) genes were identified in the 5% CO$_2$ growth condition. The top 30 differentially expressed genes based on their FDR-adjusted p-values (padj) are represented in Fig 1B. Among the 291 significantly up-regulated and down-regulated genes, *sap*, *pagR*, *pagA* and branched-chain amino acid (BCAA) genes have previously been reported to show similar transcriptional changes [10,23,24].

We performed gene set enrichment analysis (GSEA) of the entire transcriptome (5237 genes, including chromosomal and pXO1 genes) to identify key pathways altered under toxin-producing conditions (5% CO$_2$). Notably, we observed a substantial downregulation of genes involved in central carbon metabolism, particularly pyruvate metabolism and TCA cycle-related genes. Gene sets significantly enriched with padj < 0.05 are depicted in Fig 1C. Focusing on key KEGG pathways for downregulated gene-sets (bar01200, bar00020, and bar00620), we identified the leading-edge genes, such as *pckA*, *citZ*, and *sucC*, (Figs 1D, S1A), genes known to be regulated by carbon catabolite repression proteins like CcpA [25]. These findings align with previous studies demonstrating that anthrax toxin production is indirectly governed by CcpA-regulated sugar metabolism [7,9].

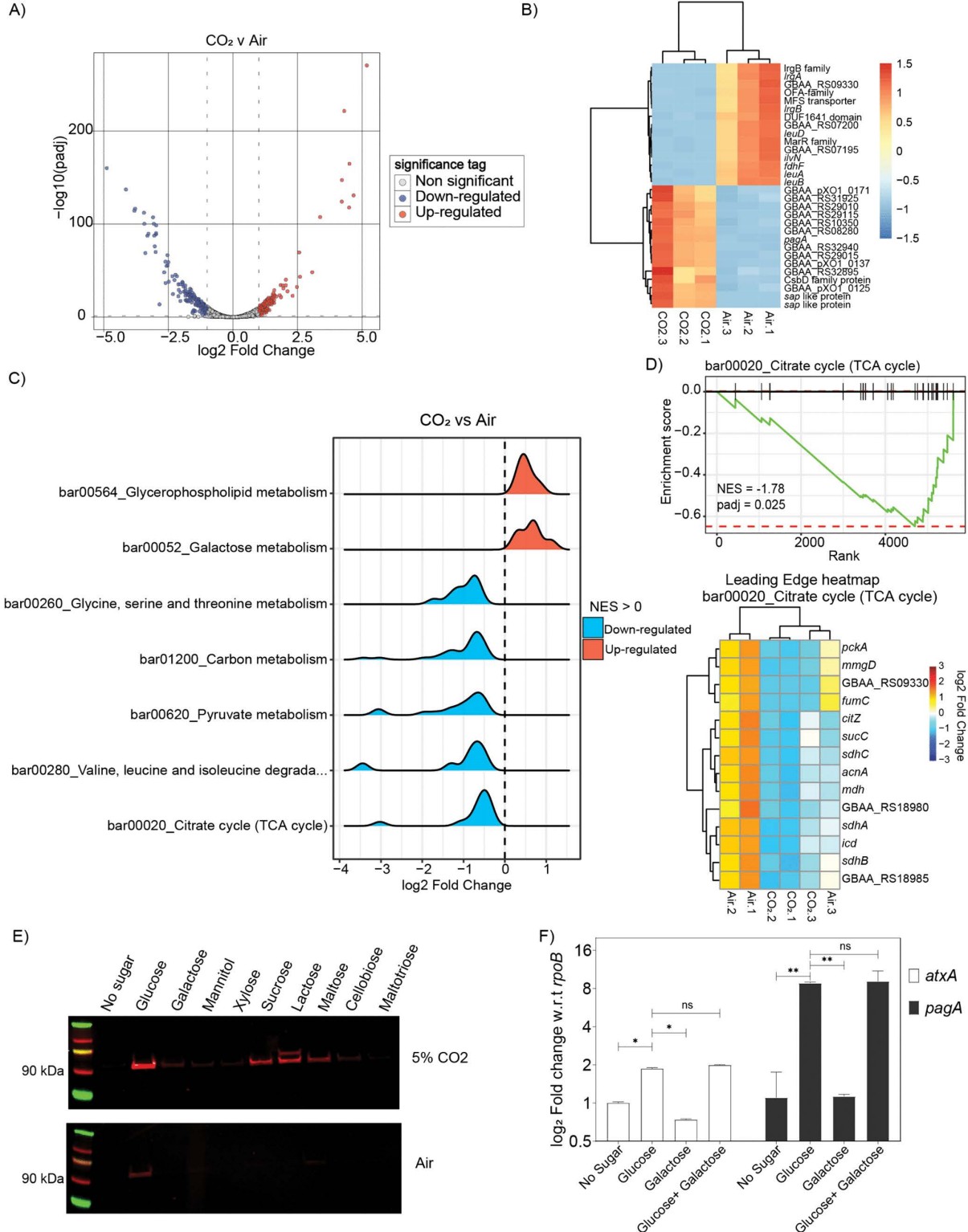

**Fig 1. A) Volcano plot of gene expression for 5237 chromosomal and pXO1 genes, with log2 fold change on the x-axis and log10 adjusted p-value on the y-axis.** Genes with adjusted p-value < 0.05 and log2 fold change > 1 are shown in red (up-regulated), while those with log2 fold change < -1 are shown in blue (down-regulated). Bacteria were grown in NBY-G (NBY + 10mM glucose) media with and without 5% $CO_2$. B) Heatmap

of the top 30 dysregulated genes (15 down-regulated and 15 up regulated) under 5% $CO_2$ condition sorted by adjusted p-values, with color gradient indicating linear fold change. Hypothetical genes are annotated by their locus tag number. Biological replicates of are labeled as Air.1 to Air.3 and $CO_2$.1 to $CO_2$.3 for bacteria grown under 5%$CO_2$. C) Ridge plots of selected metabolic pathways dysregulated in *B. anthracis* Wt under toxin-producing (10mM glucose+5% $CO_2$) vs. non-toxin-producing (10mM glucose+air) conditions. Based on Net Enrichment Score (NES), Up-regulated pathways (NES>0) are shown in red, and down-regulated pathways (NES<0) are shown in blue. D) Enrichment score profile for the TCA cycle pathway (bar00020). Vertical bars represent gene grouping based on net enrichment scores. The heatmap below shows leading genes defining the pattern in Fig 1C, with color gradient indicating linear fold change. Biological replicates of are labeled as Air.1 to Air.3 and $CO_2$.1 to $CO_2$.3 for bacteria grown under 5%$CO_2$. E) Immunoblot of PA (88kDa) expression in Wt grown with and without 5% $CO_2$ in RM minimal media supplemented with specific sugars. Representative image of n=7 experiments. Supernatant volume in each lane is normalized to total cellular protein concentration (mg/mL) estimated using BCA method. F) qRT-PCR analysis of *atxA* and *pagA* transcripts in bacteria grown with different sugars and 5% $CO_2$. Expression levels are normalized to *rpoB* transcripts. ns=non-significant, p<0.05, * p<0.01. Representative graph of n=3 experiments.

Based on the identification of central carbohydrate metabolism genes, especially glycolytic and PTS$^{glu}$ genes from the GSEA as responsive to growth in $CO_2$, we further investigated the role of fermentable sugars in anthrax toxin production. Bacteria were grown in minimal media supplemented with different carbon sources (Fig 1E). Remarkably, glucose with 5% $CO_2$ induced a~5-fold increase in the expression of the toxin component, protective antigen (PA), compared to other fermentable sugars with 5% $CO_2$ (Fig 1E). Sugars containing one unit of α-D-glucose (e.g., sucrose, lactose, maltose) also triggered toxin component expression, but only in the presence of 5% $CO_2$. Importantly, the enhanced toxin production was not solely due to improved bacterial growth because of presence of fermentable sugars; rather, the transcription of *atxA* and *pagA* was upregulated by 1.8-fold and 8-fold, respectively, under these conditions (Fig 1F) compared with the bacteria grown under no-sugar condition. It is important to note that other sugars like galactose do not have inhibitory effects on the genes regulated in presence of glucose. This suggests that glucose, combined with 5% $CO_2$ synergistically activates the toxin synthesis machinery. Supporting these observations, our GSEA revealed significant upregulation of pathways such as bar00010, bar02060, bar00620, and bar01200 - containing glycolytic and TCA cycle genes - when transcript abundance was compared between bacteria grown in the presence or absence of supplemented sugar, while 5% $CO_2$ was present in both cases (S1 Fig).

## Glycolysis and TCA intermediates are significantly altered under toxin producing conditions

Initial gene-set enrichment analyses in Fig 1 highlighted the potential for the role of central carbon metabolism, particularly glycolysis and the TCA cycle, in regulating anthrax toxin production. Specifically, a marked upregulation of glycolytic genes in the presence of glucose, contrasted with the downregulation of TCA and pyruvate metabolism genes under 5% $CO_2$ supplementation, suggests a strong positive correlation between upper glycolytic intermediates and toxin production in *B. anthracis*, while an inverse correlation is observed between lower glycolytic or TCA intermediates and toxin production. To explore the steady-state levels of glycolytic and TCA intermediates in toxin producing versus non-producing conditions, we compared their relative abundance in bacteria grown in minimal RM media [8,26] with or without 10mM glucose and 5% $CO_2$ (Fig 2A).

As shown in Fig 2A, the levels of central metabolic intermediates (FDR-corrected padj) shifted significantly in response to glucose and $CO_2$. In wild-type (Wt) bacteria, glucose levels increased by 5.7-fold, while fructose-1,6-bisphosphate (FBP) and lactate levels were elevated by 2-fold and 2.7-fold, respectively. The concentration of phosphoenolpyruvate (PEP), a key regulator of glucose uptake via the PTS$^{glu}$ system, decreased by 0.49-fold under these conditions. These results are consistent with our GSEA analysis, which showed significant enrichment of glycolysis and pyruvate metabolism pathways under 10mM glucose (S1C Fig). Additionally, genes involved in the glucose specific phosphoenolpyruvate:-carbohydrate phosphotransferase system (PTS$^{glu}$) showed a pattern of upregulation in GSEA (S1A, S1B Fig) and were enriched in the toxin-producing condition, albeit with a slightly relaxed significance threshold (padj<0.188). We identified the leading-edge genes (*PtsG*, *ptsHI* operon) (S1B Fig), which are also under direct regulation of CcpA. These could be potential genes linking CcpA-mediated toxin regulation in *B. anthracis.* PEP levels are particularly important for regulating

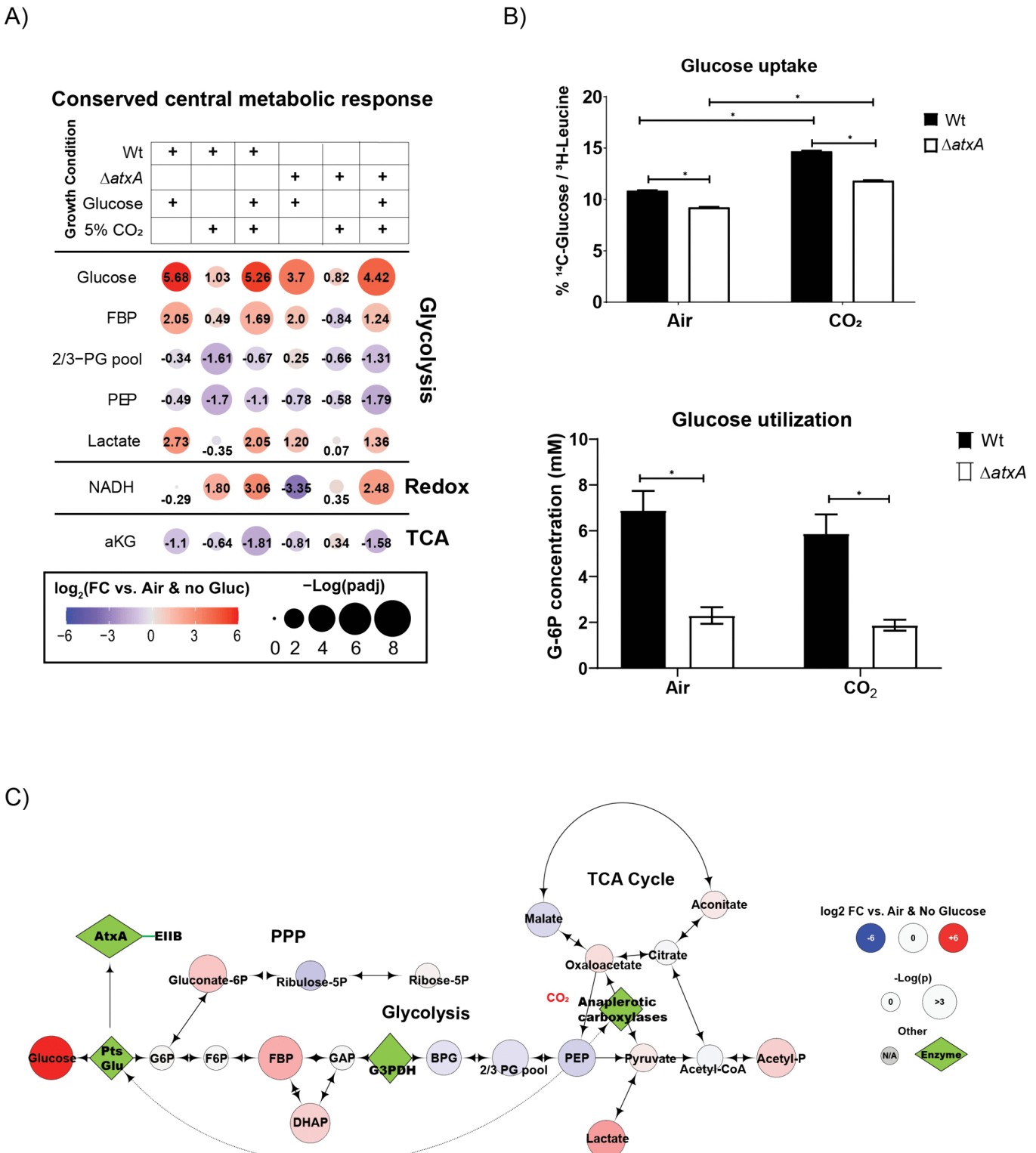

**Fig 2. A) Bubble plot showing the relative abundance of central carbon metabolism metabolites significantly dysregulated under different growth conditions in RM minimal media for *B. anthracis* Wt and Δ*atxA* strains.** The color gradient (red to blue) and the number in each bubble indicates fold change in metabolite levels in respective conditions compared to *B. anthracis* Wt. grown in air + no-glucose in RM minimal media conditions.

Bubble size represents the adjusted p-value as -log(padj). B) Upper bar graph of glucose uptake in *B. anthracis* Wt and Δ*atxA* strains grown in RM minimal media with or without 5% $CO_2$. Uptake is measured with a $^{14}$C-glucose pulse for 15 min and normalized to $^3$H-Leucine in the bacterial pellet. The lower bar graph shows glucose utilization as levels of cytosolic glucose-6-phosphate under similar conditions. Representative graph of n = 3 experiments. C) Metabolic map bubble plot (like Fig 2A) showing the relative abundance of glycolytic, PPP, and TCA intermediates and their connections. The color gradient (red to blue) indicates fold change in metabolite levels in respective conditions compared to *B. anthracis* Wt grown in air + no-glucose in RM minimal media conditions. Bubble size represents the adjusted p-value as -log(padj). Critical enzymes relevant to glycolysis, anaplerosis and *atxA* gene activation are shown as green diamonds.

glucose uptake through PTS$^{glu}$ [27], so we focused on PEP under different conditions. In Wt bacteria grown in 5% $CO_2$ (without glucose), PEP levels dropped by 1.7-fold compared to growth in air (Fig 2A). Even with 10 mM glucose and 5% $CO_2$, PEP levels remained suppressed by 1.1-fold compared to air-grown bacteria. Interestingly, this trend was consistent in the *atxA* null-mutant (Δ*atxA*), suggesting that PTS$^{glu}$ operates upstream of AtxA-dependent toxin production in *B. anthracis*.

To validate these observations, we measured glucose uptake and utilization under varying conditions. Fig 2B demonstrates that glucose uptake was significantly reduced in Δ*atxA*, with a 25% reduction in intracellular glucose compared to Wt, regardless of growth conditions. Similarly, glucose utilization was lower in Δ*atxA*, implying that AtxA either directly influences the PTS$^{glu}$ pathway or that glucose uptake and metabolism are upregulated only under toxin-producing conditions. Furthermore, our data also suggest that *B. anthracis* leverages glycolysis to generate high-energy molecules like NADH and acetyl phosphate, which may support the increased protein synthesis required for toxin production (Fig 2A, 2C).

Fig 2C presents a metabolic map comparison of glycolytic intermediates between bacteria grown in toxin-producing conditions (5% $CO_2$ + 10 mM glucose) and non-toxin-producing conditions (air + no glucose), providing a better understanding of how PEP levels may regulate toxin production in *B. anthracis*. Under toxin-inducing conditions, we observed a significant depletion of PEP, the key molecule triggering PTS$^{glu}$-dependent glucose uptake. Notably, pyruvate levels were significantly enriched under toxin-producing conditions, indicating higher PEP utilization (S2A, S2B Fig). Consistent with this, TCA intermediates (oxaloacetate, citrate, and isocitrate) were elevated, suggesting that anaplerotic carboxylases [28] - enzymes for carbon-fixation through anaplerosis are active during toxin production (S2A, S2B Fig). Based on these metabolic profiles, we hypothesize that anaplerotic flux along with PTS$^{glu}$ plays a crucial role in modulating PEP levels in toxin-producing conditions. This suggests that *B. anthracis* prioritizes glucose utilization for toxin production, even though elevated levels of glycolytic intermediates such as pyruvate (S2B Fig) can negatively regulate PTS$^{glu}$-dependent glucose uptake [29].

## AtxA physically interacts with PTS$^{glu}$ and anaplerosis proteins

To better understand the mechanism of toxin production in the presence of 10 mM glucose and 5% $CO_2$, we investigated whether AtxA, the transcription factor essential for toxin production, is directly regulated by the PTS$^{glu}$ system and/or anaplerosis. As seen in Fig 1E, omission of either glucose or 5% $CO_2$ from the growth medium results in the loss of toxin induction. Therefore, we compared the AtxA interactome in bacteria grown with glucose in the absence and presence of external 5% $CO_2$. AtxA-interacting proteins were isolated from Wt and Δ*atxA* strains overexpressing AtxA-His6 via pAMY1-*atxA* (Table 1) using co-affinity purification, as described in the methods section, followed by identification of the interactome through MS/MS analysis. Figs 3A and 3B show the absolute abundance of peptides derived from the most strongly AtxA-interacting proteins in bacteria grown in the presence (A) or absence (B) of 5% $CO_2$.

Under 5% $CO_2$, we specifically identified an interaction between AtxA and PtsG (Uniprot #A0A6L8PWR6, Fig 3A), a glucose permease in the PTS$^{glu}$ pathway. Interestingly, we also detected an interaction between AtxA and pyruvate carboxylase in $CO_2$ as well as in air (Pyc; Uniprot #A0A6L7HFM3, Fig 3A, 3B), one of the anaplerotic carboxylases in *B. anthracis*. However, the interaction of Pyc and AtxA in air questions its relation to $CO_2$-dependent regulation. Additional

**Table 1. List of recombinant plasmids and bacterial strains.**

| Name | Genotype | Reference |
|---|---|---|
| Ames 35 (Wt) | *B. anthracis str.* Ames 35 pXO1$^+$/pXO2$^-$ | |
| pAMY1 | *B. anthracis* expression plasmid with IPTG inducible system (*laO$^+$* promoter, *lacI$^+$*) | This study |
| pAMY1-*atxA* (AtxAc) | AtxA-His6 expression plasmid in *B. anthracis* | This study |
| pAMY1-*atxA* H199A | AtxA-H199A expression plasmid in *B. anthracis* | This study |
| pAMY1-*atxA* H199D | AtxA-H199D expression plasmid in *B. anthracis* | This study |
| pAMY1-*atxA* H379A | AtxA-H379A expression plasmid in *B. anthracis* | This study |
| pAMY1-*atxA* H379D | AtxA-H379D expression plasmid in *B. anthracis* | This study |
| pPROExHTC-*atxA* H199D | AtxA-H199D expression plasmid in *E. coli* | |
| pSC | Single-crossover plasmid; *ts*-replicon | [46] |
| pCrePAS | | [46] |
| pSC-*ptsG*-LF | pSC inserted with *ptsG* upstream homologous sequence | This study |
| pSC-*ptsG*-RF | pSC inserted with *ptsG* downstream homologous sequence | This study |
| *B. anthracis* Δ*ptsG* | Ames 35 (*ptsG$^-$*) | This study |
| *B. anthracis* Δ*ptsG*:: *ptsG* comp | Ames 35 (*ptsG$^-$*) with *ptsG* reconstituted at its locus | This study |
| pSC-*pyc*-LF | pSC inserted with *pyc* upstream homologous sequence | This study |
| pSC-*pyc*-RF | pSC inserted with *pyc* downstream homologous sequence | This study |
| *B. anthracis* Δ*pyc* | Ames 35 (*pyc$^-$*) | This study |
| *B. anthracis* Δ*pyc*:: *pyc* comp | Ames 35 (*pyc$^-$*) with *pyc* reconstituted at its locus | This study |
| *B. anthracis* Δ*ptsG*/ Δ*pyc* | Ames 35 (*ptsG$^-$*, *pyc$^-$*) | This study |
| *B. anthracis* Δ*atxA* | Ames 35 (*atxA$^-$*) | This study |
| *B. anthracis* Δ*ptsG*/ Δ*atxA* | Ames 35 (*ptsG$^-$*, *atxA$^-$*) | This study |
| *B. anthracis* Δ*pyc*/ Δ*atxA* | Ames 35 (*pyc$^-$*, *atxA$^-$*) | This study |
| *B. anthracis* Δ*ptsG*/ Δ*pyc*/ Δ*atxA* | Ames 35 (*ptsG$^-$*, *pyc$^-$*, *atxA$^-$*) | This study |
| *B. anthracis* BH500 | *B. anthracis* (pXO1$^-$, pXO2$^-$) with ten proteases mutant | [51] |
| pMRC4 | *B. anthracis* mini replicon carrying expression plasmid with C4 gene (a GFP variant) under *pagA* promoter | [6] |
| pMRI-C4 | pMRC4 with *pagA* promoter replaced with constitutive promoter (*lacO* fusion promoter and *lacI$^-$*) | This study |
| pMRI-*atxA*-C4 | pMRI-C4 with *atxa* gene in frame at 5' of C4 gene. | This study |
| pAMY1-*atxA*-C4 (AtxA-C4) | AtxA-C4 expression plasmid in *B. anthracis* | This study |
| pTN8 | *B. subtilis* mCherry expression plasmid | Kumaran S. Ramamurthi lab |
| pAMY1-*mCherry* | mCherry from pTN8 cloned into pAMY1 | This study |
| pAMY1-*ptsG-mCherry* | pAMY1 inserted with *ptsG-mCherry* fusion gene at multiple cloning site | This study |
| pKA2 | Plasmid containing pUB110 replicon, *Kmr* for selection both in *E. coli* and *B. anthracis*, and functionally active *atxA*. | [6] |
| pKA2-H199A | pKA2 with H199A substitution in AtxA | [6] |
| pKA2-H199D | pKA2 with H199D substitution in AtxA | [6] |
| pKA2-H379A | pKA2 with H379A substitution in AtxA | [6] |
| pKA2-H379D | pKA2 with H379D substitution in AtxA | [6] |

AtxA interaction partners are presented in S3A, S3B, and S3C Fig. Interestingly over 42 and 67 different proteins were identified in the AtxA interactome in air and $CO_2$ growth conditions, respectively (S2 File). As a transcription factor, it was expected that AtxA would interact with various proteins involved in coupled transcription-translation, regardless of the growth conditions. Over 50% of them belong to bacterial co-transcription/translation machinery.

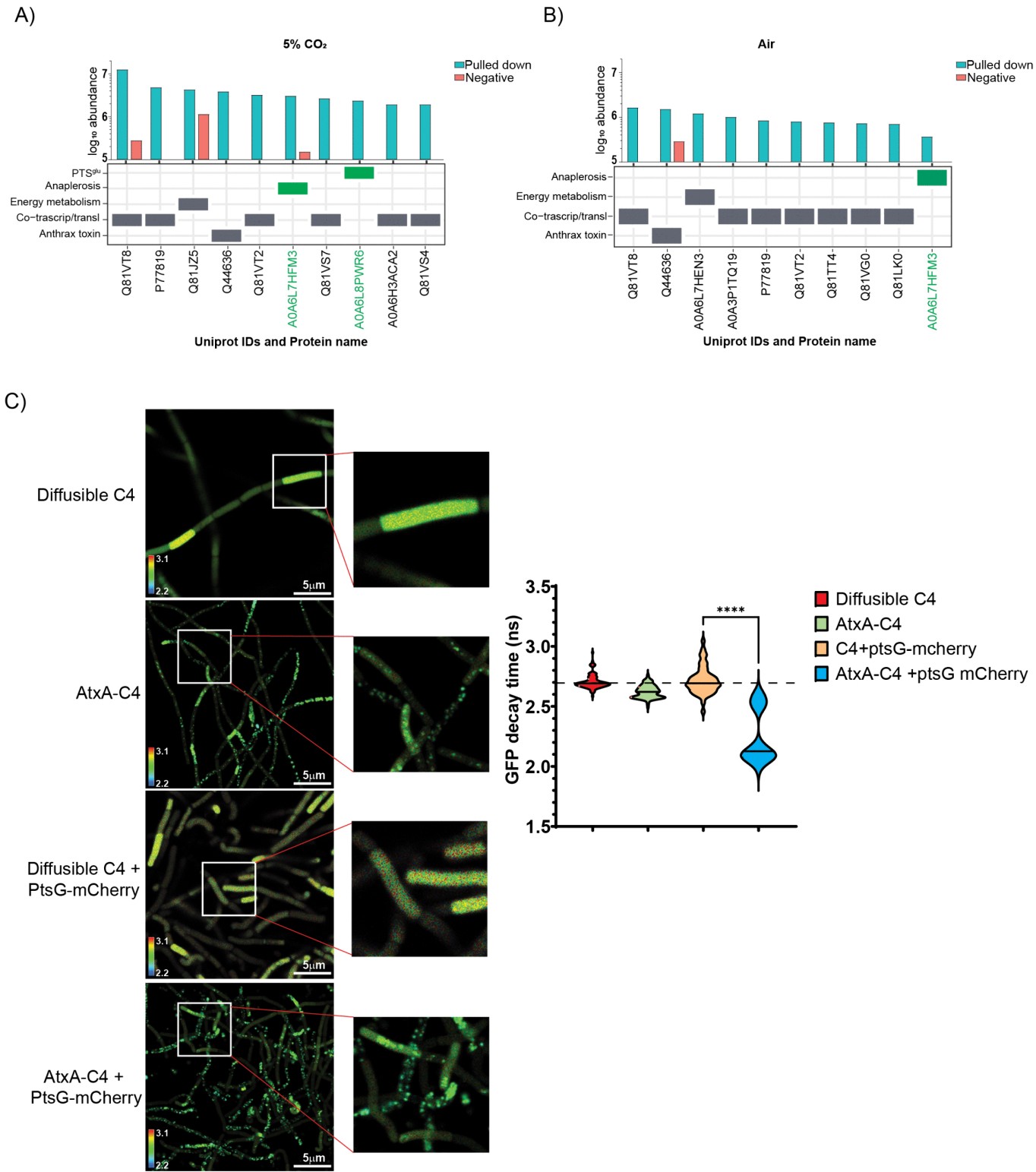

**Fig 3. A, B) Bar graphs showing the absolute abundance (y-axis) of sum of intensities of all peptides for respective proteins (Uniprot IDs, x-axis) identified from co-affinity purification samples.** Cyan bars indicate abundance in the pulled-down sample, while red bars indicate

background or non-specific proteins. Each protein is classified based on its molecular function in the table below. Proteins of interest, PtsG (A0A6L8PWR6) and Pyc (A0A6L7HFM3), are marked with green boxes. A) AtxA interactions identified from bacteria grown in $CO_2$. B) AtxA interactions identified from bacteria grown in air. Glucose was present in both conditions. C) Representative FRET-FLIM micrographs of strains expressing different FRET partners performed in NBY-G medium supplemented with 0.8% $NaHCO_3$ and incubated under 5% $CO_2$. The inset shows high-resolution pixelated images of each bacillus in the chain. The violin/distribution plot represents the median GFP decay profile (in nanoseconds, ns) of the GFP variant C4 in the respective strains. The graph represents $n = 5$ experiments, with each sample counting for $n > 100$ bacilli. ****indicates p-value $< 0.001$. Scale bar $= 5\mu m$.

To further investigate how PtsG might regulate AtxA function, we assessed the interaction between these proteins using Fluorescence Lifetime Imaging Microscopy (FLIM) in the presence of 5% $CO_2$. We chose strains expressing AtxA-C4 and PtsG-mCherry fusion proteins for these experiments. The lifetime decay profile of the GFP variant C4 is shown in Fig 3C. Notably, the lifetime decay of AtxA-C4 was significantly reduced from 2.7 nano-second to 1.95 nano-second when PtsG-mCherry was co-expressed, indicating a 28% faster decay of GFP fluorescence. This faster decay suggests a strong physical interaction between AtxA and PtsG within the bacterial cell. Additionally, we observed a slight shift toward a faster decay profile for AtxA-C4 when compared to the diffusible C4 variant, likely due to homo-FRET between GFP units of AtxA-C4 dimers [30]. These observations confirm a physical interaction between PTS$^{glu}$ and AtxA, suggesting that PTS$^{glu}$ may influence AtxA dimerization and activation, potentially modulating its role in toxin production in *B. anthracis*.

## PtsG and Pyc are essential for anthrax toxin production

Building on the observation that AtxA binds PtsG, we sought to determine the roles of PtsG as well as Pyc in anthrax toxin production. We generated null mutants of these genes (ΔptsG, Δpyc, and ΔptsG/Δpyc) in *B. anthracis* Ames 35 and tested their ability to produce the toxin components PA, LF, and EF under minimal medium conditions, with regulated amount of fermentable sugar (glucose) in the media.

As shown in Fig 4A, the ΔptsG mutant exhibited a nearly 90% reduction in the expression of all toxin components compared to Wt. The ΔatxA strain served as a negative control, as AtxA is essential for toxin production [10]. Similarly, the Δpyc mutant showed significantly reduced levels of PA and LF, though not to the extent observed in the ΔptsG strain. Notably, the double mutant (ΔptsG/Δpyc) completely abolished toxin component expression (Fig 4A). This loss of toxin production in the ΔptsG strain was fully reversible when complemented with a single copy of the gene reinserted into the *Bacillus* genome (S4A Fig). Despite PtsG being a critical glucose transporter in *B. anthracis*, the ΔptsG mutants displayed growth patterns like Wt, with only minor, non-significant growth defects (S4B Fig).

We measured the levels of key metabolites, including the lower glycolytic intermediates (particularly PEP and pyruvate) and TCA intermediates (citrate and oxaloacetate), in the ΔptsG, Δpyc, and ΔptsG/Δpyc strains under toxin-producing conditions in RM minimal media (S4C Fig). Consistent with our observations in Fig 2, the levels of lower glycolytic intermediates were significantly reduced in the ΔptsG and ΔptsG/Δpyc strains, highlighting the importance of PEP in anthrax toxin production. Interestingly, the levels of upper glycolytic intermediates were significantly elevated in these mutants, suggesting enhanced gluconeogenesis and a subsequent depletion of PEP and PTS signaling. Additionally, the Δpyc strain showed significantly elevated pyruvate and lactate levels, confirming the loss of anaplerotic activity in these strains.

Since previous studies have reported that AtxA's transcriptional activity depends on histidine phosphorylation [21,22], we investigated whether this post-translational modification could restore AtxA function in the absence of upstream PTS$^{glu}$ and anaplerotic signals. To test this, we generated ΔptsG and Δpyc mutants in the ΔatxA background and complemented them with phospho-mimetic and phospho-ablative mutants of AtxA at histidine 199 and 379 (H199D, H379D, H199A, and

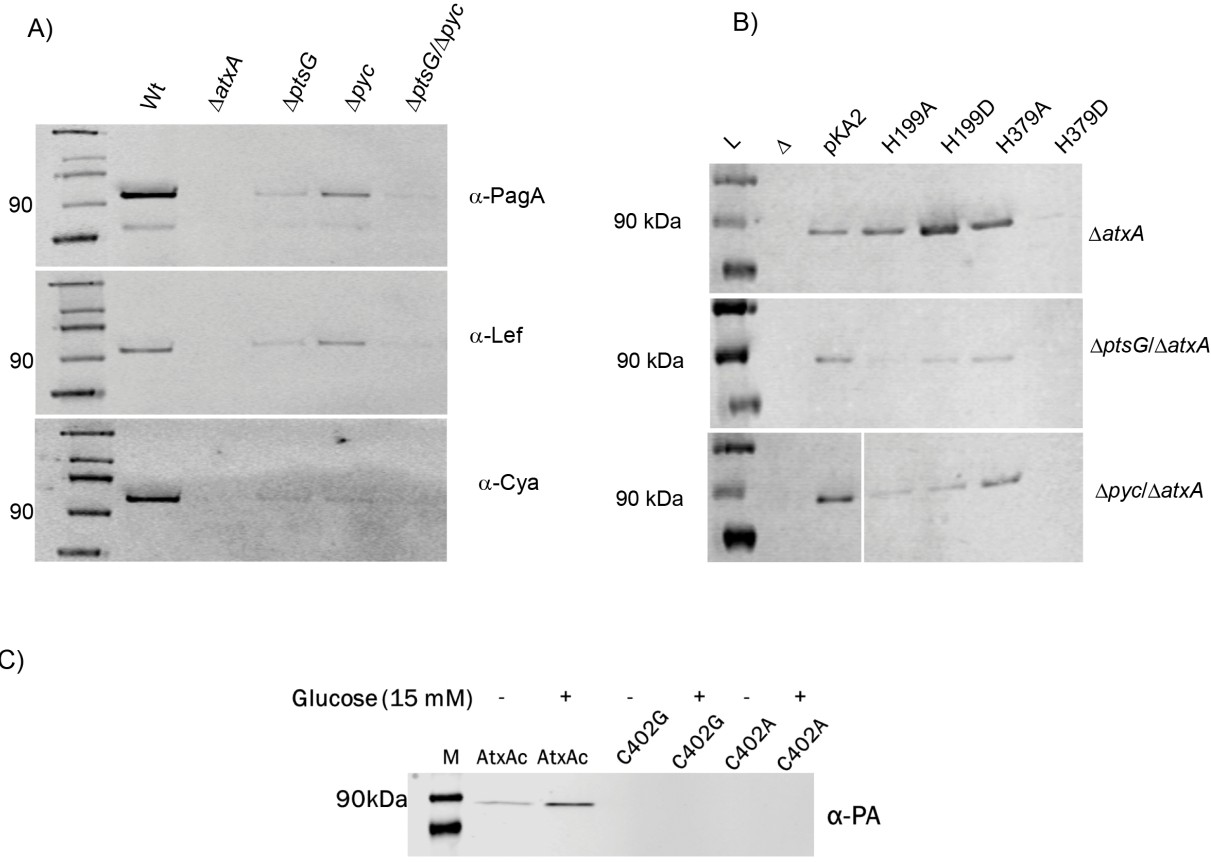

**Fig 4. Immunoblotting for anthrax toxin components from different PTS$^{glu}$ and Pyc mutants. A) Immunoblotting for anthrax toxin components (PA, LF, and EF) in *B. anthracis* Wt and PTS$^{glu}$ and Pyc mutants.** Supernatant volumes were normalized against total cellular protein for comparative analysis. B) Effect of AtxA phosphomimetic (H199D and H379D) and phosphoablative (H199A and H379A) mutants on PA production in *B. anthracis* Δ*ptsG* and Δ*pyc* strains, in the background of Δ*atxA* strains. pKA2 expresses Wt-AtxA; H199A expresses the AtxA-H199A mutant protein; H199D expresses the AtxA-H199D mutant protein; H379A expresses the AtxA-H379A mutant protein; and H379D expresses the AtxA-H379D mutant protein. C) Effects of C402 mutations in the AtxA-EIIB domain on PA expression. Having AtxA variants expressed in the background of *B. anthracis* Δ*atxA* strains, with plasmid pKA2 expressing the respective AtxA variants.

H379A, respectively). As previously reported [22], we found that the H199D and H379D mutations had opposing effects on AtxA's transcriptional activity, as measured by PA expression in the culture supernatant. However, these phenotypes were not restored in the Δ*ptsG*/Δ*atxA* or Δ*pyc*/Δ*atxA* strains (Fig 4B), further indicating that the interactions between AtxA and PtsG/Pyc are critical for AtxA's transcriptional function.

To explore how the physical interaction between AtxA and PtsG might influence AtxA's transcriptional activity, we examined the structure of AtxA's EIIB domain and identified a conserved CIXR motif (residues 402–405). This CIXR motif forms a similar pocket to that found in homologous transcription factors such as MtlR, which binds inorganic phosphate during the phosphorelay in the PTS$^{glu}$ signaling pathway. We hypothesized that this CIXR motif is crucial for AtxA's interaction with the PTS$^{glu}$ pathway, contributing to its functional activity. Indeed, mutations in this motif (C402A and C402G) completely abolished AtxA's transcriptional activity in *B. anthracis* (Fig 4C). The role of EIIB domain was further supported by FLIM assay where interaction between AtxA-ΔEIIB-C4 and PtsG-mCherry resulted in GFP decay profile like AtxA-C4 alone with lifetime decay of 3.05 nano-second (S4D Fig). This suggests that the EIIB domain of

AtxA is essential for in-cell interaction of AtxA with PtsG, with its deletion resulting in loss of anthrax toxin expression (S4E Fig). Surprisingly, the lifetime decay of AtxA-ΔEIIB-C4 (2.8 nano seconds) was 10% lower than AtxA-C4 (3.08 nano seconds) suggesting its role in regulating AtxA dimerization. Loss of EIIB may cause the AtxA-ΔEIIB-C4 protein to multimerize or aggregate [31] inside cell resulting in stronger homo-FRET within aggregated C4, thus a faster lifetime decay profile.

## Loss of PtsG and Pyc attenuates *B. anthracis virulence*

Given the significant reduction in anthrax toxin production observed in the Δ*ptsG* mutants (both single and double mutants with Δ*pyc*), we sought to determine whether these effects would translate into reduced bacterial virulence and pathogenicity in an actual infection scenario. Since the primary role of PtsG is to import glucose into bacteria, we assumed that under limited nutrient conditions within the host environment, the Δ*ptsG* and Δ*ptsG*/Δ*pyc* mutants would exhibit reduced germination or growth [1]. Therefore, to understand the role of *ptsG* and *pyc* in regulating anthrax toxin production in animals, we bypassed the early stages of *B. anthracis* infection by subcutaneously injecting $1 \times 10^5$ vegetative bacteria into C57BL/6J mice. As shown in Fig 5A, the Δ*ptsG*/Δ*pyc* double mutant was completely attenuated, with 10 out of 15 mice surviving until 168 h post-infection, in stark contrast to mice infected with the Wt strain, which all succumbed to infection by 96 h post-infection. The single Δ*ptsG* mutant also showed significantly reduced virulence, with 7 out of 15 mice surviving to 168 h post-infection. The attenuated phenotype of the Δ*ptsG* mutant was completely reversed in the *ptsG* complemented strain (*ptsG*-Comp, Fig 5A). These results suggest that PTS$^{glu}$ acts as a critical sensor to initiate toxin production and establish infection. Interestingly, the single Δ*pyc* mutant did not show a major reduction in virulence, indicating that *B. anthracis* may be utilizing an alternative pathway to regulate PEP levels, thus maintaining PTS$^{glu}$-dependent toxin production.

Furthermore, when $8 \times 10^6$ spores of the Δ*ptsG* and Δ*ptsG*/Δ*pyc* strains were injected subcutaneously, both mutants were completely attenuated, demonstrating a severe defect in their ability to establish and spread infection (S5 Fig). To validate these findings, we measured the dissemination of the Δ*ptsG* mutant in the host post-infection. As illustrated in Fig 5B, the Δ*ptsG* mutant showed approximately a 10-fold reduction in the number of bacteria disseminating to host organs by 48 h post-infection, further confirming the attenuated behavior of these mutants, likely due to the lack of toxin production.

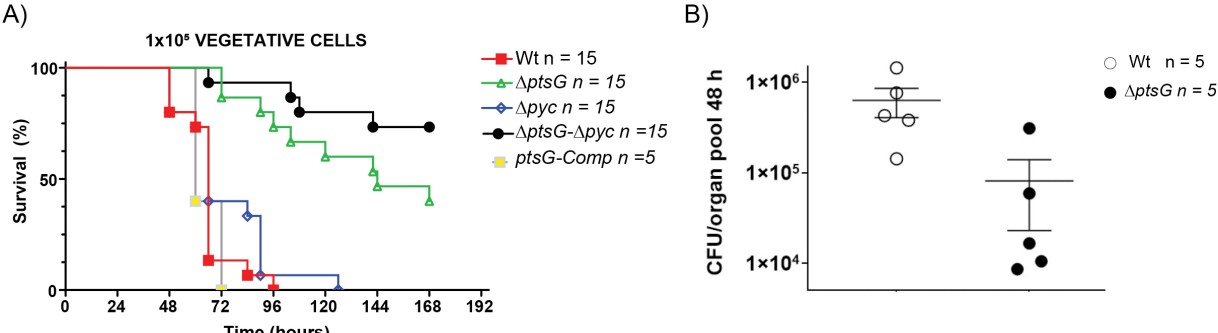

**Fig 5. A) Virulence of parent and PTS$^{glu}$ and Pyc mutants. A. Survival curves of mice infected sub-cutaneously (s.c.) with vegetative *B. anthracis* are shown.** C57BL/6J mice were injected s.c. with $1 \times 10^5$ CFU of the Wt (red line; n = 15), Δ*ptsG* (green; n = 15), Δ*pyc* (blue; n = 15), Δ*ptsG*/Δ*pyc* (black; n = 15), and Δ*ptsG*-Comp (yellow; n = 5) **p < 0.05. B) CFU/g$^{-1}$ of tissue collected. Two sample permutation analysis was performed to compare each strain to the other. **p < 0.01.

## Discussion

The growth and progression of disease caused by *B. anthracis* are influenced by its metabolic functions and virulence factors [32,33], with both types of activities likely being synchronized during infection. The phosphoenolpyruvate:carbohydrate phosphotransferase systems (PTSs), which are bacterial sugar transport mechanisms, might play a role in this synchronization. These systems facilitate nutrient uptake, regulate overall metabolism, and influence virulence in a variety of pathogens [34]. PTS systems facilitate the uptake of sugars via sugar-specific permeases known as enzymes II, which include the cytoplasmic domains EIIA and EIIB, as well as the membrane-spanning domain EIIC (and occasionally EIID). During the transport process, sugars are phosphorylated by phosphoenolpyruvate (PEP) through intermediary phosphate carriers such as the general enzyme I and HPr. The phosphorylation state of PTS components is influenced by the availability of sugar substrates and the fluxes of glycolysis/gluconeogenesis, which together determine the PEP/pyruvate ratio. Consequently, PTS phosphorylation mirrors the overall metabolic state, a feature that many bacteria utilize as a sensory mechanism to regulate metabolism, respiration, motility, and virulence. However, the processes by which the specific phosphor-transfer reactions carried out by individual sugar-specific PTS branches or components are incorporated into a bacterial virulence strategy to regulate its pathogenesis are still not well understood.

In this study, we revealed a crucial synergistic role of the glucose transport protein PtsG and a metabolic regulator Pyc in *B. anthracis* virulence (Fig 6). The long-standing role of glucose and $CO_2$ in growth media is sought to be essential for anthrax toxin production and therefore its virulence. Our study correlates with the "Jail-break" model of anthrax pathogenesis [1], which signifies the importance of the host systemic system to support *B. anthracis* growth and proliferation during pathogenesis.

As shown in Fig 6, we propose that these changes are coordinately regulated by PtsG and Pyc to activate AtxA, the master virulence regulator, and therefore anthrax toxin production and its virulence. The host lymph and blood are rich sources of both glucose and $CO_2$, which may trigger metabolic changes in the bacteria. A previous study [24] identified PtsG and BioY among the top 25 significantly induced genes when *B. anthracis* is grown in mammalian blood, suggesting a strong correlation of glucose and anaplerotic genes in anthrax biology, especially toxin production. The PtsG gene encodes the first component of $PTS^{glu}$ which senses the presence of free glucose in the environment, like host blood or lymph. BioY encodes for a biotin transporter involved in the uptake of biotin, which is essential for the function of anaplerotic carboxylases like Pyc. We showed that once the bacteria sense the presence of glucose and $CO_2$ together in the growth environment, it immediately (within 15 min of glucose supplementation) activates $PTS^{glu}$ to take up the glucose. This triggers glycolysis to generate a pool of PEP, which initiates the phosphotransferase activity in the $PTS^{glu}$ to keep taking more glucose into the cell. Since a higher amount of fermentable sugar can build byproducts like pyruvate/lactate and acetate in the bacteria, it uses another metabolic regulator, Pyc, to activate the anaplerotic pathway, which uses external $CO_2$ to pivot off the byproduct of PEP (pyruvate) so that bacteria could keep making more PEP and utilize it for the PTS machinery [28]. The transcription factor AtxA, with its EIIB domain, physically interacts with PtsG and gets activated, possibly by its phosphorylation (from unknown mechanism) and dimerization during the phosphotransferase activity in PTS. The activated AtxA then regulates the expression of toxin genes and virulence in *B. anthracis*.

Using a global transcriptional profiling approach, we identified a set of 291 genes that were significantly dysregulated in bacteria under toxin-producing conditions. Using a similar approach, we previously identified 318 differentially regulated genes [10]. The discrepancy in the number of genes identified in the two studies could be attributed to differences in bacterial growth stages (OD600 1.8 vs 2 in the previous study) or the use of different analysis tools for RNA mapping and differential gene expression. Nonetheless, given the similar levels of differentially expressed genes, we used our dataset to understand the global changes that occur in bacteria when grown in the absence and presence of 5% $CO_2$ and how these changes correlate with anthrax toxin expression. Our data supports the hypothesis that once the bacteria sense the presence of glucose and $CO_2$ together in the growth environment, they immediately modulate the central carbon metabolism pathways (bar_01200), the TCA cycle (bar_00020), and glucose-specific PTS (bar02060) (Fig 1). There are several

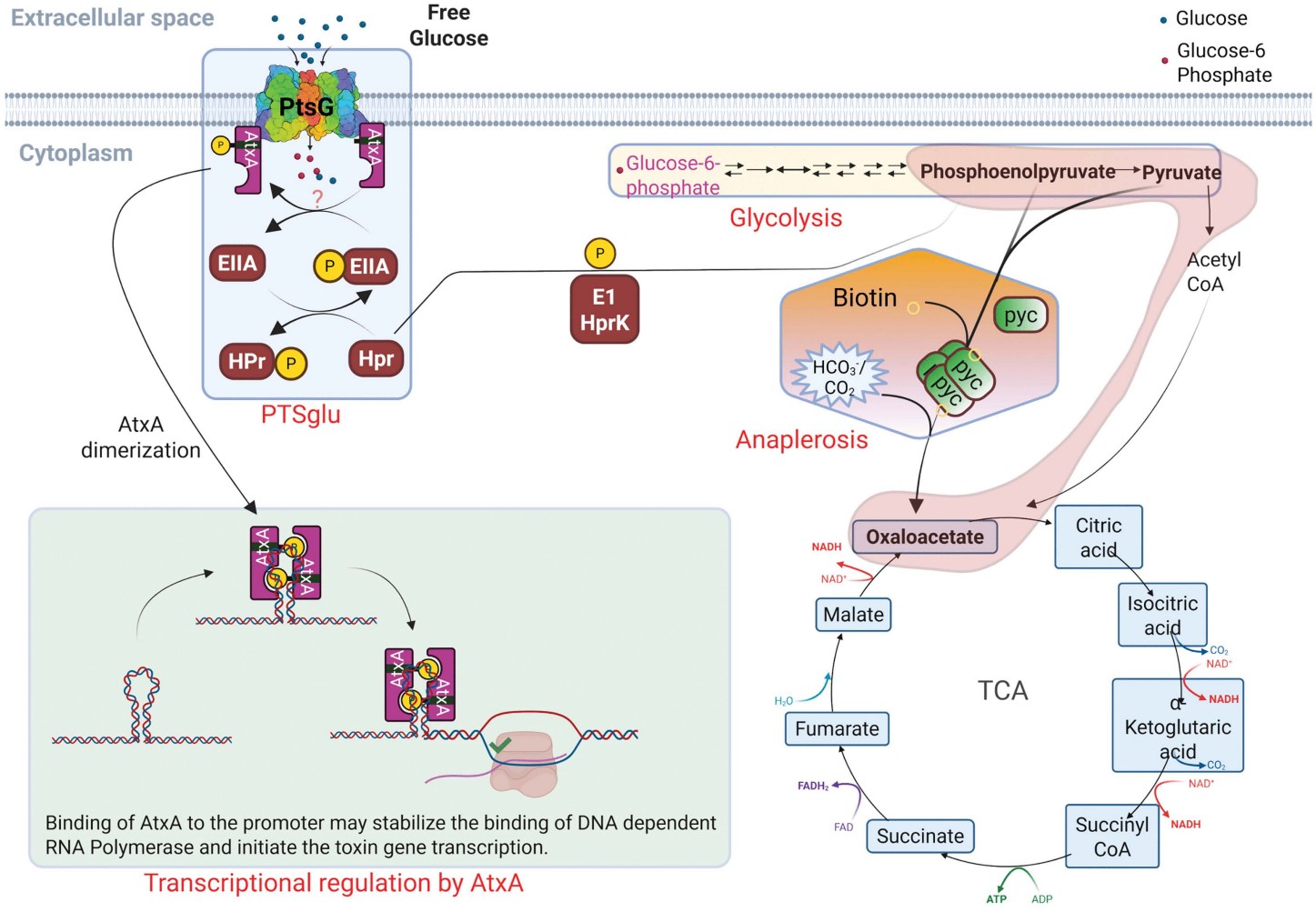

**Fig 6. Model describing the synergistic regulation of anthrax toxin production by glucose (PTS$^{glu}$) and CO$_2$ (anaplerosis) metabolic pathways through modulation of AtxA transcriptional activity in *B. anthracis*.** Created in BioRender. Bothra, A. (2026) https://BioRender.com/zub2ejq.

reports suggesting the role of sugar transport and sugar metabolism genes which all are under control of central carbon metabolism, to regulate anthrax toxin production and its pathogenesis [7,9,10,21].

As has been shown previously, the utilization of fermentable sugars like glucose will accumulate fructose-1,6-bisphosphate, an agonist of carbon catabolite repression (CCR), and drive the bacteria to utilize glycolysis to generate high-energy molecules [35]. Other Firmicutes like *S. pyogenes* and *B. subtilis* have well-described PTS systems where the PTS components are under the influence of CCR and are known to regulate similar pathways as we discovered in *B. anthracis* [15,36]. This is validated by our metabolic profile of *B. anthracis* under toxin-producing conditions (glucose and 5% CO$_2$, Fig 2), where we clearly observed that bacteria generate all the glycolytic intermediates; however, the Pentose-phosphate-pathway (PPP) and Tri-carboxylic acid (TCA) intermediates were significantly lower under these conditions. The observation was biochemically validated where we observed that *B. anthracis* takes up more glucose and the rate of glucose utilization is higher under toxin-producing conditions. Changes in the abundance of PEP and pyruvate, the key regulators of PTS, and the relative expression of anthrax toxin components under the toxin-producing conditions suggest a strong association of PTS$^{glu}$ with the activation of AtxA. The metabolic profile of Δ*ptsG* and Δ*ptsG*/Δ*pyc* also

validates our hypothesis as these mutants show a poor abundance of PEP, indicative of reduced glucose uptake and PTS flux, further supporting PtsG as the principal glucose permease linking carbon metabolism to toxin regulation. Also, the loss of *pyc* (Δ*pyc*) impacts the bacteria to start accumulating pyruvate and lactate suggesting the loss of anaplerotic pathway in these mutants (S4 Fig).

However, it is not clear from our results which PTS component could result in AtxA phosphorylation, but our protein interaction data strongly suggest that AtxA does physically interact with PtsG in the cell (Fig 3). Using a co-affinity purification approach we clearly identify the major interacting partners of AtxA under different growth conditions. Being a transcription factor, it is intuitive to identify several transcriptional and translational proteins to physically interact with AtxA. Such breadth of interacting partners is consistent with prior affinity-based studies of bacterial transcription regulators, which frequently co-purify with components of the transcription-translation machinery and metabolic partners [13,15,37]. Specific pulldown of PtsG and Pyc as AtxA interactors under toxin-producing conditions validates our hypothesis that PTS$^{glu}$ and anthrax toxin expression are coupled. The interaction between PtsG and AtxA was further validated with the FRET-FLIM assay. There is an exponential correlation between the decay profile and distance between the FRET partners [38]. Based on this correlation, a 28% faster decay of the GFP lifetime signal suggests a strong interaction between AtxA and PtsG with a possible distance of 22–28Å. However, this interaction was completely lost with the ΔEIIB domain of AtxA, suggesting a crucial role in interaction with PtsG and therefore its activation (S4D, S4E Fig). However, it is difficult to understand a faster lifetime decay in AtxA-ΔEIIB-C4 alone. One possible explanation for this phenotype could be an aggregation of AtxA-ΔEIIB-C4 in cell resulting in homo-FRET in C4 and thus a faster decay profile. Further, we identified the importance of C402 in the EIIB domain of AtxA, which possibly helps in AtxA dimerization as reported earlier [22,31]. However, based on the alpha-fold prediction [39] of dimeric AtxA (S5A Fig) with the AtxA binding element of the *pagA* promoter [40], the C402 of one unit of dimeric AtxA directly sits on top of the H379 of the other unit of the AtxA dimer with a possible distance of 10.2 Å, and thus may be involved in some electrostatic interactions (S5B Fig). Since the interaction volume of a phosphate ion can range up to 20 Å^3, the H379 phosphorylation may affect these interactions, resulting in AtxA de-dimerization. Interestingly, analogous regulatory modifications have been described for other PRD/EIIB-like transcriptional regulators: a cysteine residue within the EIIB-like domain of the Mga regulator in *Streptococcus pyogenes* was proposed to undergo phosphorylation [41], and the *Bacillus subtilis* regulator MtlR has been shown to phosphorylate at C419 in the EIIB(Gat)-like domain [42]. These parallels raise the possibility that C402 of AtxA may likewise be subject to redox- or phosphate-mediated modification that modulates its activity.

Complementation with phosphomimetic (H199D and H379D) and phosphoablative (H199A and H379A) mutants of AtxA also showed that the PtsG and AtxA interaction is a prerequisite for its activation irrespective of whether AtxA is phosphorylated or not. To the best of our knowledge, the critical interaction of PTS proteins with PRD-domain containing transcription factors for their activation has never been shown before. As a result of the loss of PTS$^{glu}$ function and anaplerotic (Pyc) function (Fig 5), the Δ*ptsG*/Δ*pyc* double mutant completely lost virulence post-infection in mice. Given that attenuation persisted even when vegetative bacteria were used for infection (Fig 5A) and that all mutants exhibited near-normal growth in culture (S4B Fig), the reduced virulence most likely reflects a defect in infection establishment rather than an early germination or growth limitation. These strains possibly could not sense the presence of free glucose and $CO_2$ in the host systemic system and thus could not activate the bacteria to produce enough toxin to suppress the host immune system and establish the infection. The hypothesis seems correct as spores of these mutants are completely attenuated in a mice model of infection (S5C Fig). Also, the dissemination rate of these mutants is extremely slow post-infection.

Our results demonstrated a direct interaction between AtxA and PtsG complement earlier findings by Bier et al. [7], which proposed that PTS$^{glu}$ components may influence *atxA* gene expression indirectly through unknown regulators. Together, these observations suggest that both transcriptional and post-translational mechanisms contribute to AtxA regulation in *B. anthracis*, providing a more integrated view of how carbon-source sensing connects to toxin control.

Thus, based on these findings, we finally propose AtxA as an integral component of the PTS$^{glu}$, where the transcriptional activity of AtxA is regulated by environmental signals like glucose and $CO_2$. The major limitation of our study is to understand how PTS$^{glu}$ and/or Pyc could affect AtxA phosphorylation, however, it has been previously known that other components of *B. anthracis* PTS machinery do not have a direct role in AtxA phosphorylation, in in vitro conditions. There could be several issues with these assays possibly because of the labile nature of histidine phosphorylation [7]. Nonetheless, our data strongly supports our model suggesting the role of environmental signals, like glucose and free $CO_2$, in the host systemic system as a key regulator of anthrax toxin production and pathogenesis. This study reveals how extracellular bacteria like *B. anthracis* adapts its metabolic pathways to survive and regulate virulence under diverse environmental conditions.

## Materials and methods

### Bacterial Strains, Growth, and Media Conditions

Bacterial growth conditions were used as described previously with modifications [43]. The *Escherichia coli* strain DH5α was used for cloning purposes. For *B. anthracis* transformations, plasmids were first passed through *E. coli* SCS110 to obtain unmethylated DNA. *E. coli* and *B. anthracis* strains were grown in Luria Bertani (LB) broth or on LB agar plates (Difco). The *B. anthracis* Ames 35 (pXO1$^+$, pXO2$^-$; Wt) strain was grown in a modified Nutrient Broth Yeast Extract (NBY) medium, composed of 0.8% nutrient broth and 0.3% yeast extract, supplemented in some cases with 10 mM glucose (NBY-G). For selection, ampicillin (100 µg/mL), kanamycin (25 µg/mL), and erythromycin (400 µg/mL) were used in *E. coli*, while kanamycin (10 µg/mL) and erythromycin (10 µg/mL) were used for *B. anthracis*. All cultures were incubated at 37°C with proper aeration (using a 1:5 headspace ratio) and shaking at 200 rpm. For toxin synthesis, *B. anthracis* strains were grown in NBY medium supplemented with 0.8% sodium bicarbonate (NaHCO₃) and incubated for 4 h at 37°C in the presence of external 5% $CO_2$. To assess growth kinetics, glycerol stocks of *B. anthracis* strains were streaked on LB agar plates, and an isolated colony was used to inoculate primary cultures. Log-phase bacterial cultures were then used to initiate secondary cultures at a starting optical density at 600 nm (A600) of 0.02. Growth was monitored by measuring A600 every 2 h for up to 10 h. For metabolomics experiments, strains were grown in RM minimal medium [8,26], with (RM-G) or without (RM) 10 mM glucose supplementation. For toxin production, the medium was supplemented with 0.8% NaHCO₃, and cultures were incubated for 4 h at 37°C under 5% $CO_2$. A detailed description of plasmids and strains used in this study is provided in Table 1.

### Cloning and protein expression

All plasmids used in this study were created using NEB HiFi methods (New England Biolabs). Briefly, homologous sequences of 19–22 nucleotides were used to generate homologous pairs between the insert and the vector to create individual clones. All plasmids used in *B. anthracis* strains were first passed through the *dam$^-$/dcm$^-$ E. coli* strain SCS110. Primers and gBlocks used for all molecular work are listed in Tables 2, 3.

For the AtxA co-affinity purification assay, the *atxA* gene with a C-terminal His6 tag was cloned in frame under the IPTG-inducible P*$_{spac}$* promoter [44] to generate the pAMY1-*atxA* clone. Site-directed mutagenesis was used to create H199A, H199D, H379A, and H379D mutants in pAMY1-*atxA*.

For the FLIM assay, a two-plasmid system was used as described previously [45]. Briefly, pAMY1-*ptsG*-mCherry was created using PCR amplicons of the *ptsG* gene and mCherry derived from the *B. anthracis* genome and pTN8-mCherry plasmid (Kumaran S. Ramamurthi lab, NIH). The products contained homologous sequences between pAMY1 and the amplicon to generate pAMY1-*ptsG-mCherry* with the *ptsG* gene in frame under the IPTG-inducible promoter and an in-frame C-terminal *mCherry* gene. Similarly, pMR [6] was used to generate pMRI-*atxA*-C4 with the IPTG-inducible *atxA* gene and a C-terminal GFP variant (C4).

**Table 2. List of DNA oligonucleotide/primers used in this study.**

| Name | Sequence | Remarks | Reference |
|---|---|---|---|
| *atxA*-RT-F | GGAAGAGAACAAGATGTCCGCG | qRT-PCR primer | This study |
| *atxA*-RT-R | GTTATAGCGAACAACACACACAATTTG | qRT-PCR primer | This study |
| *pagA*-RT-F | AATGAATCAGAATCAAGTTCC | qRT-PCR primer | This study |
| *pagA*-RT-R | ATGTATATTCATCACTCTTCTTAAC | qRT-PCR primer | This study |
| *rpoB*-RT-F | ACTGCTACTGTAATTCCAAACCGCGG | qRT-PCR primer | This study |
| *rpoB*-RT-R | AAACCCTAATGCGCGTAACAAAACAG | qRT-PCR primer | This study |
| *atxA*-Comp-F | CAGCTAGCAAAGGAGAACGTATATGCTAACACCGATATCCAT | Cloning primer (with RBS) | This study |
| *atxA*-Comp-R | TTGCATGCTTAATGATGGTGGTGGTGATGCCCTTGAAAATATA-AATTTTCTCCGGATCCTATTATCTTTTTGATTTCATGAAAATCTC | Cloning primer (carry TEV and HIS6 sequence) | This study |
| *atxA*-C402G-SDM-F | TCACAAGGATGGTTACATCGAGAG | Mutagenesis primer | This study |
| *atxA*-C402A-SDM-F | TCACAAGGATGCTTACATCGAGAG | Mutagenesis primer | This study |
| *atxA*-C402-SDM-R | GGTGTATACATATATTTTTTTCG | Mutagenesis primer | This study |
| *atxA*-ΔEIIB-C4-F | GAAAATTTATATTTTCAAGGGGTTACACG | Mutagenesis primer | This study |
| *atxA*-ΔEIIB-C4-R | CATACGTTGAGTTTCAAAATGCATT | Mutagenesis primer | This study |
| C4-For | CACCACCATCATTTAGCATGCATGACTGCTTTAACTGAAG | Cloning primer | This study |
| C4-Rev | CCACCGAATTAGCTTGCATGCTCATTGATAAGTATCTAAATCCAC | Cloning primer | This study |
| mCherry-For | GGTACCATGGTGAGCAAGGGCGAG | Cloning primer | This study |
| mCherry-Rev | AAGCGGCCGCTTTACTTGTACAGCTCGTCCATGC | Cloning primer | This study |
| pSC-*ptsG*-LF-For | GGCTCGAGAATTGCTGAAGGTGTTGTAGATCTTTTAAATTCT | Cloning primer | This study |
| pSC-*ptsG*-LF-Rev | GGACTAGTGGAAGCATTAACGCTTTTCCGAC | Cloning primer | This study |
| pSC-*ptsG*-RF-For | GGCTCGAGGGGGAGACGGATTTGCAATTGAG | Cloning primer | This study |
| pSC-*ptsG*-RF-Rev | GGACTAGTGCAGTGTTTACAAGTAGAGTTGCTG | Cloning primer | This study |
| pSC-*pyc*-LF-For | ACGTCTCGAGAACAGATTTTATTATGGCAATTATATCT-GAAGAACTAGGT | Cloning primer | This study |
| pSC-*pyc*-LF-Rev | ACGTGAGCTCACCACTATCCTCTTTGGAATAGATTGCAACT | Cloning primer | This study |
| pSC-*pyc*-RF-For | ACGTCTCGAGAATCGGAGAGCTTCAGCCAGA | Cloning primer | This study |
| pSC-*pyc*-RF-Rev | ACGTGAGCTCAGTACGTCACGAACGAGCTAGC | Cloning primer | This study |

**Table 3. List of gBlocks used in the study.**

| Name | Reference |
|---|---|
| pSC-*ptsG* Comp | This study |
| pSC-*pyc* Comp | This study |
| IPTG-RBS-C4 | This study |
| RBS-*mCherry* | This study |

RBS = ribosome binding site

## Null mutants and complementation generation

Single-gene knockout strains for *ptsG* and *pyc*, as well as the double knockout strain *ptsG/pyc*, were generated in *B. anthracis* Ames 35 (Δ*ptsG*, Δ*pyc*, and Δ*ptsG*/Δ*pyc*) and in the *B. anthracis* Δ*atxA* background (Δ*ptsG*/Δ*atxA* and Δ*pyc*/Δ*atxA*) using the method previously described [46]. Briefly, two single-crossover plasmids derived from the temperature-sensitive shuttle vector pSC were used sequentially to insert *loxP* sites flanking the regions of interest in the *ptsG* and *pyc* genes. A third temperature-sensitive plasmid, pCrePAS [46], expressing Cre recombinase, was then employed to excise the DNA

region between the *loxP* sites, resulting in the deletion of the target gene from the *B. anthracis* Ames 35 genome. Positive colonies were confirmed by PCR using primers flanking the gene and internal gene-specific primers.

For the complemented strain (Δ*ptsG::ptsG*), the *ptsG* gene along with 374 bp of upstream sequence was PCR-amplified using primers that introduced 5' PstI and 3' XbaI restriction sites. The amplified product was digested with PstI and XbaI and ligated into the pSC-*ptsG*-LFLR shuttle vector, which had previously been used for generating the Δ*ptsG* strain. Positive clones were confirmed by sequencing, and unmethylated plasmid DNA was electroporated into the Δ*ptsG* strain. Transformants were selected on LB agar plates supplemented with erythromycin (10 µg/mL). Once *ptsG* was reinserted at its native locus, pCrePAS was used again to excise the plasmid DNA between the *loxP* sites. Colony PCR was performed for confirmation using *ptsG*-specific primers. The details of the primers used in this study are listed in Table 2.

## Total RNA isolation

For total RNA isolation from *B. anthracis*, pellets from 5-ml bacterial cultures with average OD600 of 1.8 were resuspended in 1 mL TRIzol, and 250 µL of 0.1 mm zirconia beads (BioSpec Products) were added. The cells were homogenized using a FastPrep-24 homogenizer (MP Biomedicals) at a speed of 6.0 m/s for 60 sec, followed by a 5-min incubation on ice. This process was repeated for three rounds of homogenization. RNA was then isolated as previously described [47]. Briefly, the aqueous phase from the TRIzol-lysed cells was treated with 0.5 M LiCl and 0.7 volumes of isopropanol, followed by a 2-h incubation on ice to precipitate the RNA. The samples were centrifuged at 13,000 rpm for 30 min at 4°C, and the resulting RNA pellet was washed twice with 75% nuclease-free ethanol. The air-dried RNA was resuspended in 100 µL of nuclease-free water and stored at -80°C for future use.

## Quantitative real-time polymerase chain reaction (qPCR)

qPCR was done as described previously with modifications [47]. The isolated RNA was treated with TURBO DNase I (Thermo-Fisher Scientific) to remove any residual DNA and then purified using the RNeasy Mini Kit (Qiagen). One microgram of purified RNA was used to synthesize cDNA with the iScript cDNA Synthesis Kit (Bio-Rad). qPCR was performed using 5 ng of cDNA as the template per reaction in the QuantStudio 7 Flex Real-Time PCR System (Thermo-Fisher Scientific), with gene-specific primers listed in Table 2.

## Transcriptomics and Gene set enrichment analysis

All work described in this manuscript was conducted using *B. anthracis* Ames 35 (NC_007530) carrying the pXO1 plasmid (NC_007322) as the parent strain. Total RNA was isolated from *B. anthracis* strains grown under the various conditions outlined in the Results section and assessed for quality using an RNA TapeStation (Agilent). Samples with RNA integrity numbers (RIN) greater than 7.3 were selected for whole transcriptome analysis. The selected RNA samples were sent to Azenta Life Sciences for transcriptomic sequencing and analysis.

After library preparation and sequencing, reads were trimmed to remove adapter sequences and low-quality nucleotides using Trimmomatic v.0.36. The cleaned reads were then mapped to the reference genomes available in ENSEMBL using the Bowtie2 aligner v.2.2.6, and BAM files were generated. Unique gene hit counts were calculated using featureCounts from the Subread package v.1.5.2, and only unique reads aligning within gene regions were counted. The gene hit counts were then used for downstream differential expression analysis. Differential gene expression analysis was performed using DESeq2, comparing gene expression between the defined groups of samples. The Wald test was used to generate p-values and log2 fold changes. Genes with an adjusted p-value < 0.05 and an absolute log2 fold change > 1 were considered differentially expressed. Principal component analysis (PCA) was also performed to assess similarities and differences within and between the sample groups.

Significantly impacted pathways were identified using gene set enrichment analyses (GSEA) performed using fgsea [48]. Pathway gene sets for *B. anthracis* AMES ancestor (KEGG Organism_code: bar) were obtained from KEGG and log2 fold change from differential gene expression analysis was used as the ranking metric. A threshold of adjusted p-value (padj) < 0.05 was set to identify significantly enriched gene sets. Plots summarizing pathway analysis were created with tidyverse and cowplot packages in R.

## Metabolite extraction and data analysis

Metabolite sample preparation and data analysis was done as described earlier [49]. For all liquid chromatography-mass spectrometry (LCMS) methods, LCMS grade solvents were used. Tributylamine and all synthetic molecular references were purchased from Millipore Sigma. LCMS grade water, methanol, isopropanol, and acetic acid were purchased through Fisher Scientific.

*B. anthracis* strains were grown in RM minimal media with or without glucose and 5% $CO_2$ as mentioned in the manuscript at 37°C to ~$9 \times 10^7$ cells/ml before collection at 8,000 x g for 10 min at 4°C. Cells were washed twice with HEPES-NaCl buffer and pellets were flash frozen in dry-ice and ethanol and stored at -80°C until processing. To process cells for metabolomic analysis, cell pellets were thawed on ice, resuspended in 150 µL ice-cold methanol (Sigma) and then incubated at room temperature for 10 min. Following incubation an equal volume of LCMS grade water was added, and the samples were vigorously vortexed, then centrifuged at 13,000 x g for 15 min. Supernatants were collected, filtered in a 0.2 µM nitrocellulose syringe filter (GE Healthcare), and then diluted 1:3 before analysis. All samples were separated using a SCIEX ExionLC AC system and measured using a SCIEX 5500 QTRAP mass spectrometer. Polar metabolites were analyzed using a previously established ion pairing method with modification [50]. Quality control samples were injected after every 10 injections to control for signal stability. Samples were separated with a Waters Atlantis T3 column (100Å, 3 µm, 3 mm X 100 mm) using a binary gradient from 5 mM tributylamine, 5 mM acetic acid in 2% isopropanol, 5% methanol, 93% water (v/v) to 100% isopropanol over 15 min. Each metabolite was identified and measured with two ion fragmentation pairs and a defined retention time.

All signals were integrated using MultiQuant Software 3.0.3. Signals with greater than 50% missing values were discarded and remaining missing values were replaced with the lowest registered signal value. All signals with a QC coefficient of variance greater than 30% were discarded. Metabolites with multiple ion pairs were quantified with the signature that displayed the highest signal to noise. All filtered datasets normalized against the total signal sum for the injection prior to analysis. Single and multi-variate analyses were performed in MarkerView Software 1.3.1. All univariate comparisons were subjected to a Benjamini-Hochberg cut-off at a false discovery rate of 5%.

## Glucose uptake and utilization assays

For glucose uptake assay bacteria was grown in RM minimal media without glucose in presence and absence of 5% $CO_2$ for 3hr till they at 37°C. Strains were grown overnight in 2 mL of LB medium at 37°C under aerobic conditions. The following day, cultures were centrifuged at 3,500g for 6 min at room temperature, washed once with NBY-G medium, and resuspended in fresh NBY-G. The cultures were then reinoculated in 5 mL of NBY-G or NBY medium and grown either in the presence of 5% $CO_2$ (with 0.8% $NaHCO_3$ in the medium) or in air (with 50 mM MOPS in the medium). Cultures were grown until reaching an OD600 of 0.8, after which they were transferred to the Radioactivity lab incubator, preheated to 37°C and 5% $CO_2$. To each culture, 1 µCi of [³H]-leucine was added and the cultures were incubated for an additional 30 min. Subsequently, 0.5 µCi of [¹⁴C]-2-deoxyglucose (prepared in 15 mM non-radioactive glucose) was added to each culture. Cultures grown in air were covered with non-breathable tape to minimize $CO_2$ exchange. Samples were collected at different time points (15, 30, and 60 min). At each time point, 1 mL of culture was centrifuged at 13,000g for 6 min at room temperature, washed once with 1X PBS, and resuspended in 100 µL of 1X PBS. This suspension was then added to a scintillation cocktail for the measurement of CPM and DPM values.

## Whole cell lysate preparation and co-affinity purification and interactome analysis

*B. anthracis* strains were cultured in 200 mL of NBY-G medium with and without 5% $CO_2$. The strains used included Δ*atxA*::pAMY1 and Δ*atxA*::pAMY1-AtxA-TEV-His6. Positive controls for Ni-NTA purification consisted of lysates from the Δ*atxA*::pAMY1 strain spiked with E. coli-purified AtxA-His6, as well as purified AtxA-His6 in buffer for recovery control. Once cultures reached an OD600 of ~ 0.2, 50 μM IPTG was added, and growth continued until the cultures reached an OD600 of ~ 2. Cells were harvested by centrifugation at 6,200g for 10 min at 4°C, and the wet weights were recorded. Pellets were snap-frozen on dry ice and stored overnight. The following day, the cells were thawed in an ice-water bath and resuspended in 50 mL of binding buffer (5 mM imidazole, 0.5 M NaCl, 5 mM β-mercaptoethanol, 20 mM Tris pH 7.2) supplemented with an EDTA-free protease inhibitor cocktail. Cells were lysed by two rounds of French press, and the soluble fraction was collected by centrifugation at 10,000g for 10 min at 4°C.

For co-purification, the soluble material was clarified by an additional centrifugation at 18,000 rpm for 10 min at 4°C, followed by a 20-min incubation at 37°C. In the meantime, 1 mL of Ni-NTA resin was equilibrated with 50 mL of binding buffer for 20 min. The equilibrated Ni-NTA resin was then added to the samples, and the mixture was incubated at 4°C for 2 h to allow His-tagged proteins to bind to the resin. Following incubation, the resin was transferred to a column and washed at room temperature by gravity flow using the following steps- three rounds of 5 mL binding buffer, two rounds of 5 mL Wash Buffer 1 (40 mM imidazole pH 7.9, 1.0 M NaCl, 20 mM Tris pH 7.2, 5 mM β-mercaptoethanol), one round of 10 mL Wash Buffer 1, one wash with 1.5 mL High Salt Wash Buffer (40 mM imidazole pH 7.9, 1.5 M NaCl, 20 mM Tris pH 7.2, 5 mM β-mercaptoethanol), and three rounds of 2.5 mL Wash Buffer 1. His-tagged AtxA and its interacting proteins were eluted in three 500 μL aliquots of elution buffer (20 mM Tris pH 7.2, 800 mM imidazole pH 7.9, 500 mM NaCl, 5 mM β-mercaptoethanol). The eluted samples were analyzed by SDS-PAGE and sent to the core facility for protein identification via mass spectrometry.

## Analysis of anthrax toxin proteins-PA, LF and EF

*Bacillus* strains were grown in NBY broth, and cells were harvested by centrifugation at 12,000g. The cell pellets were resuspended in 1X PBS. Cells were lysed using bead-beating with 0.1 mm Zirconium beads (MP-Biomedicals) using MP-FastPrep24 with a setting of 6 M/Sec for 60 seconds. The lysate was rested on ice for 5 min, and the lysis was repeated for three cycles. Lysate was cleared of cellular debris by centrifuging at 13000 rpm for 30 min at 4°C. Protein concentration in the whole-cell lysates was determined using the BCA Protein Assay Kit (Thermo-Fisher Scientific) and used to assess toxin protein synthesis. The culture supernatant was filtered through a 0.22 μm filter assembly, and proteins were precipitated with 20% trichloroacetic acid (TCA). The precipitated proteins were resuspended in resuspension buffer [1% SDS, 10% glycerol, 10 mM Tris-HCl (pH 6.8), 2 mM EDTA, 100 mM DTT, and 1X protease inhibitor cocktail (Roche Applied Science)] and used to evaluate the secretion of PA, LF, and EF proteins. The volume of the supernatant was normalized to the total protein content estimated from the whole-cell lysate using the BCA assay. Immunoblot analysis was performed using primary antibodies raised in rabbit- anti-PA (1:5,000), anti-LF (1:5,000), and anti-EF (1:5,000).

## FRET-FLIM sample preparation and analysis for in-vivo interaction

For in-vivo interaction of AtxA and PtsG, FRET-measured by FLIM was performed with *B. anthracis* strain BH500 [51]. The BH500 strain is cured of pXO1 and pXO2, which provides us flexibility to use multiple compatible plasmids in background of *B. anthracis* genome. Plasmids used for FLIM assay are listed in Table 1. Single colony of BH500 strains expressing fluorescently tagged proteins (C4, AtxA-C4 and PstG-mCherry) were used to synchronize the culture growth in NBY-G medium supplemented with 0.8% NaHCO₃ and incubated under 5% $CO_2$. Once cultures reached an OD600 of ~ 0.2, 50 μM IPTG was added, and growth continued until the cultures reached an OD600 of ~ 2. Cells or samples mounted in imaging buffer were imaged using Leica-DMI8- SP8 WLL FLIM-Falcon as described before [52]. Briefly, coverslips were then mounted onto glass slides, and images were acquired using a Leica SP8WLL-FLIM falcon inverted confocal microscope with a 63 × oil immersion

objective (Leica Microsystems, Buffalo Grove IL). For fluorescence resonance energy transfer by fluorescence lifetime imaging (FRET-FLIM) analysis, fluorescence decays were acquired and resolved by time-correlated single-photon counting using SP8 FLIM FALCON system equipped with a tunable white light laser system set at 488 nm excitation wavelength at 80 MHz frequency. Images were acquired at 512–512-pixel format, collecting over 1,000 photons per pixel. FRET efficiency transients and FRET-FLIM Images were analyzed and processed using LASX Single Molecule detection analysis software (Leica Microsystems, Buffalo Grove IL). The GFP lifetime decay profiles were plotted using GraphPad Prim.

## C57BL/6J mice infection

C57BL/6 J mice (in-house, male, 8 weeks old) were used for all the infection study. Mice were either subcutaneously (SC) infected with 1 x $10^5$ CFU/ml of vegetative bacteria or with 8 x $10^6$ CFU/ml of spores of *B. anthracis* strains prepared in PBS (1 mL) in the scruff of neck and monitored daily for signs of malaise prior to euthanasia and total organ (spleen, liver and kidney) harvest. All animal experiments were performed in strict accordance with guidelines from the NIH and the Animal Welfare Act, under protocols approved by the Animal Care and Use Committee of the National Institute of Allergy and Infectious Diseases, National Institutes of Health (protocol LPD9E).

## Statistical analysis

Unless mentioned otherwise, a minimum of three independent experiments (biological) repeated thrice (technical) were done to ensure data reproducibility. GraphPad Software (Prism 6) was used to plot the data and significance of the results was analyzed using two-tailed Student's t test or one-way ANOVA followed by a post hoc test (Tukey test). p-values $< 0.05$ were considered as statistically significant.

## Supporting information

**S1 Fig. A) Ridge plots representing significantly dysregulated metabolic pathways in *B. anthracis* Wt grown under toxin-producing conditions (glucose + 5% CO$_2$) versus non-toxin-producing conditions (glucose + air).** Up-regulated pathways (NES > 0) are colored red, while down-regulated pathways (NES < 0) are colored blue. B) Representative enrichment score for the PTS pathways (bar02060_PTS). Vertical bars represent the grouping pattern of individual genes of the PTS pathway based on their net enrichment scores. The heatmap below indicates the leading genes identified based on their enrichment scores. C) Ridge plots representing significantly dysregulated metabolic pathways in *B. anthracis* Wt grown under toxin-producing conditions (glucose + 5% CO$_2$) versus non-toxin-producing conditions (no glucose + 5% CO$_2$). Up-regulated pathways (NES > 0) are colored red, while down-regulated pathways (NES < 0) are colored blue. D) Representative enrichment score for the PTS pathways (bar02060_PTS). Vertical bars represent the grouping pattern of individual genes of the PTS pathway based on their net enrichment scores. The heatmap below indicates the leading genes identified based on their enrichment scores.
(DOCX)

**S2 Fig. A) Bubble plot showing the relative abundance of divergent carbon metabolism metabolites significantly dysregulated under different growth conditions in *B. anthracis* Wt and Δ*atxA*.** The color gradient (red to blue) indicates fold change in metabolite levels compared to air and no-glucose conditions. Bubble size represents the p-value as -log(p). B) Scatter plot of selected glycolysis (black dots) and TCA (grey dots) intermediates comparing several growth conditions. Growth conditions for each comparison are mentioned at the top of each graph. The comparisons are made as follows: Variable-1 vs Variable-2 (constant). The x-axis indicates fold change in metabolite levels and y-axis represents the p-value as -log(p).
(DOCX)

**S3 Fig. A) Coomassie-stained polyacrylamide gel of the elutes from co-affinity samples.** Lanes are marked from 1-7 for the list of strains and growth conditions used in the table below. M = Protein molecular ladder. B,C) Bar graphs showing the absolute abundance (y-axis) of peptides for respective proteins (Uniprot IDs, x-axis) identified from co-affinity purification samples. Cyan bars indicate abundance in the pulled-down sample, while red bars indicate background or non-specific proteins. Each protein is classified based on its molecular function in the table below. B) AtxA interactions identified from bacteria grown in air. C) AtxA interactions identified from bacteria grown in 5% $CO_2$. Glucose was present in both conditions. (DOCX)

**S4 Fig. A) Immunoblotting for anthrax toxin components (PagA and Lef) in *B. anthracis* WT, Δ*ptsG*, and its complemented strain (Δ*ptsG*-Comp).** Supernatant volumes were normalized against total cellular protein for comparative analysis. Representative graph of n = 2 experiments. B) Growth kinetics of all *B. anthracis* strains under toxin-producing conditions (glucose + 5% $CO_2$). Each data point represents the average of three replicates. Representative image of n = 2 experiments. C) Heatmap showing the relative abundance of significantly altered metabolites in Δ*ptsG*, Δ*pyc*, and Δ*ptsG*/Δ*pyc* mutants compared to their respective abundance in *B. anthracis* WT strain grown under toxin-producing conditions (glucose + 5% $CO_2$). The left heatmap shows the log2 fold change in abundance as a color gradient from blue (low abundance) to red (high abundance). The right heatmap corresponds to the p-value of each data point on the left heatmap, represented as a color gradient from grey (p > 0.1, insignificant) to orange (p = 0-0.01, highly significant). D) Representative FRET-FLIM based interactions in strains expressing different FRET partners. The violin/distribution plot represents the median GFP decay profile (in nanoseconds, ns) GFP variant C4 in the respective strains. The graph represents n = 2 experiments, * indicates p-value. E) Immunoblotting for PagA showing effects of loss of the EIIB domain of AtxA (AtxA-ΔEIIB-C4) on anthrax toxin expression. Having AtxA variants expressed in the background of *B. anthracis* Δ*atxA* strains, with plasmid pAMY expressing the respective AtxA variants. Representative image of n = 3 experiments. (DOCX)

**S5 Fig. A, B) Alpha-fold predicted structure of WT AtxA dimer with the *pagA* promoter carrying AtxA-binding elements.** The two units of the AtxA dimer are shown in rainbow and golden colors. B) Zoomed image of the interaction between C402 of one unit (rainbow-colored) and H379 of the other unit (golden-colored). The electrostatic interaction zones of both residues are shown as white clouds around them. C) Virulence of parent and PTS$^{glu}$ and Pyc mutants. A. Survival curves of mice infected sub-cuteneously (s.c.) with spores *B. anthracis* are shown. C57BL/6J mice were injected s.c. with $8 \times 10^6$ CFU of the Wt (red line; n = 5), Δ*ptsG* (green; n = 5), and Δ*ptsG*/ Δ*pyc* (black; n = 5). (DOCX)

**S1 File. List of 291 significant (padj<0.05) differentially expressed genes in *Bacillus anthracis*, presented as log$_2$ fold change values.** Gene expression was compared between bacteria grown in NBY-G medium without $NaHCO_3$ under air conditions and those grown in NBY-G medium with 0.8% $NaHCO_3$ under 5% $CO_2$. (XLSX)

**S2 File. List of co-purified proteins identified as AtxA-interacting partners from *Bacillus anthracis* cultures grown in NBY-G medium with 0.8% $NaHCO_3$ under 5% $CO_2$ or in air without $NaHCO_3$.** Proteins were identified by mass spectrometry following co-affinity purification of AtxA-His$_6$. (XLSX)

## Acknowledgments

We thank NIAID Research Technologies Branch (RTB) staff for their support with mass-spectroscopy and microscopy. We thank Dr. Kumaran Ramamurthi (National Cancer Institute/NIH) and Dr. Kevin McIver (University of Maryland) for providing pTN8 and pKSM879 plasmids, respectively.

## Author contributions

**Conceptualization:** Ankur Bothra.

**Data curation:** Ankur Bothra, Benjamin Schwarz.

**Formal analysis:** Ankur Bothra, Benjamin Schwarz, Anupam Mondal, Mahtab Moayeri.

**Funding acquisition:** Stephen H. Leppla.

**Investigation:** Ankur Bothra, Andrei Pomerantsev, Benjamin Schwarz, Eric Bohrnsen, Nitika Sangwan, Kaitlin A. Stromberg, Mahtab Moayeri, Qian Ma, Rasem Fattah, Sundar Ganesan.

**Methodology:** Ankur Bothra, Benjamin Schwarz.

**Resources:** Catharine M. Bosio, Stephen H. Leppla.

**Supervision:** Ankur Bothra, Stephen H. Leppla.

**Validation:** Ankur Bothra.

**Visualization:** Ankur Bothra, Benjamin Schwarz, Anupam Mondal, Sundar Ganesan.

**Writing – original draft:** Ankur Bothra.

**Writing – review & editing:** Ankur Bothra, Benjamin Schwarz, Mahtab Moayeri, Stephen H. Leppla.

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
