## [Decision Letter · Decision Letter 0]

4 Nov 2025

PPATHOGENS-D-25-02437

Environmental Regulation of Toxin Production in Bacillus anthracis

PLOS Pathogens

Dear Dr. Bothra,

Thank you for submitting your manuscript to PLOS Pathogens. After careful consideration, we feel that it has merit but does not fully meet PLOS Pathogens's publication criteria as it currently stands. Therefore, we invite you to submit a revised version of the manuscript that addresses the points raised during the review process.

We look forward to receiving your revised manuscript.

Kind regards,

Bruce A. McClane

Academic Editor

PLOS Pathogens

Michael Wessels

Section Editor

PLOS Pathogens

Sumita Bhaduri-McIntosh

Editor-in-Chief

PLOS Pathogens

orcid.org/0000-0003-2946-9497

Michael Malim

Editor-in-Chief

PLOS Pathogens

orcid.org/0000-0002-7699-2064

**Additional Editor Comments:**

Your manuscript has been reviewed by three experts They each agreed that the work was significant for the field, performed at a high standard, and represented considerable effort. However, as detailed in their reviews, they did request that you consider some modifications, either to correct minor errors or to broaden the discussion to consider issues such as whether PtsG might contribute to virulence by impacting in vivo growth as well as toxin production or how conclusions from this study might relate to earlier work (e.g., work by Bier et al published in 2020). It does not appear that suitable revision will require additional experiments. Therefore, please consider and respond to the Reviewer suggestions, as listed in their reports.

**Journal Requirements:**

Potential Copyright Issues: 

i) Figure 6. Please confirm whether you drew the images / clip-art within the figure panels by hand. If you did not draw the images, please provide (a) a link to the source of the images or icons and their license / terms of use; or (b) written permission from the copyright holder to publish the images or icons under our CC BY 4.0 license. Alternatively, you may replace the images with open source alternatives. See these open source resources you may use to replace images / clip-art:

**Reviewers' Comments:**

Reviewer's Responses to Questions

**Part I - Summary**

Reviewer #1: In PPathogens-D-25-025437, the authors investigate the molecular mechanisms behind how CO2, glucose, and the AtxA regulator are linked to toxin production in B. anthracis. AtxA is known to be essential for toxin gene expression, contain PTS regulatory domains (PRDs) that can be phosphorylated on conserved histidine residues through an unknown mechanism, and have a C-terminal EIIB domain that is required for dimerization and activity. Recent work from Bier et al (2020) found that PTS EI/Hpr-activated promoter expression of AtxA, but not PRD phosphorylation; however, they couldn’t identify what PTS components might be involved. The authors have previously looked at toxin production conditions (CO2) on the B. anthracis transcriptome and initiate this study with an in-depth RNA-seq analysis and metabolomics of the combined CO2 + glucose conditions to show synergism on AtxA-dependent toxin production and metabolic transitions. They find that ∆atxA leads to reduced glucose uptake, suggesting a novel connection, and mutation in the conserved C-terminal cysteine leads to inactivity. However, deletion the PTSglu (∆ptsG) reduces toxin gene expression and attenuation in a mouse model. The authors use co-purification/pull down and FRET/FLIM to support a direct interaction between AtxA (mainly the EIIB domain) and the PTS glucose component PtsG. Finally, the use a pyruvate carboxylase (∆pyc) mutant to support their hypothesis that PEP levels drive PTS-AtxA interactions to enhance activity (ie, toxin production and virulence). The propose that AtxA is part of the PTS glu circuit, thus leading to the influence of glucose and CO2.

This manuscript represents a significant amount of work (RNAseq, metabolomics, mutational analysis, FRET/FLIM, animal studies) to address what is still an important but unanswered question not only for AtxA toxin regulation, but also for a number of homologous PRD-containing regulators in G+ pathogens…mainly, how does the PTS influence the PRD regulator based on environmental conditions. To date, direct interactions between PTS components and these regulators have been speculated but never demonstrated. So, the apparent interaction between AtxA and PtsG is significant and novel. The same can be said for AtxA acting within the PTS-Glu circuit as a means of having CO2 and glucose regulate toxin production. Thus, this work has broader impact outside of B. anthracis. Overall, the study addresses an important knowledge gap, the conclusions are supported by the data presented and the experiments well planned. Some minor comments are noted.

Reviewer #2: In this manuscript entitled “Environmental Regulation of Toxin Production in Bacillus anthracis”, Bothra and colleagues confirm a synergistic effect of glucose and CO2 on AtxA-dependent toxin production. Then they demonstrated that deletion of AtxA reduced glucose uptake in the cell, suggesting that AtxA may act within the glucose-PTS system, and AtxA interacts with the PTS glucose permease PtsG. They showed that residue C402 located in the EIIB domain of AtxA is essential for its interaction with PtsG and PtsG is required for full expression of toxin component and full virulence in a mouse model of infection. Finally, the authors propose AtxA as an integral component of the PTSglu. The study is well conducted with correct controls and bring new information on the link between environmental factors and toxin production in B. anthracis.

Reviewer #3: The manuscript by Bothra et al examines the environmental signals and molecular mechanisms that govern the activity of AtxA, a multidomain transcription factor that regulates toxin production and capsule synthesis of the bacterial pathogen Bacillus anthracis. Glucose and carbon dioxide have long been known to influence toxin production, but the direct method of regulation of these signals with regards to AtxA activity and the expression of the toxin genes has remained unclear. AtxA is classified as a PCVR [PTS-regulator domain (PRD)-containing pathogen virulence regulator] with its activity regulated by phosphorylation of His residues within the two AtxA PRDs. AtxA also contains a putative EIIB domain characteristic of PTS sugar transport EII complexes. In this work, the authors provide data supporting a synergy between glucose metabolism and anaplerosis for the induction of toxin production as well as provide evidence for a physical interaction between AtxA and the glucose-specific PTS component PtsG that is critical for bacterial virulence.

The identification of the direct link between sugar metabolism, anaplerosis, and AtxA activity is a significant step forward in current understanding of toxin regulation. The current manuscript is a tour de force in terms of the number of methods and approaches used to decipher the model presented in Fig. 6. Overall, the data provided are convincing, although I have some questions (noted below) with regards to the virulence defect of the ptsG deletion strain in mice. It would seem that the authors are indicating that the principal role for PtsG is its role in toxin induction, however it’s not clear to me how one can completely separate toxin induction from a possible role for PtsG in bacterial growth and carbon metabolism within the host. Might both be important?

**Part II – Major Issues: Key Experiments Required for Acceptance**

Reviewer #1: None

Reviewer #2: (No Response)

Reviewer #3: Specific comments:

1. Might it be possible that the AtxA EIIB C204 site is phosphorylated? This possibility was recently proposed (although not directly demonstrated) for the Streptococcus pyogenes Mga transcriptional regulator, which also contains two PRDs and a EIIB-like domain with a key C residue that appears to link carbon metabolism with Mga-regulated virulence gene expression (Woo et al, Microbiology Spectrum 2024).

2. Starting line 277: Regarding the attenuation of the ptsG deletion strains – while I appreciate that the subcutaneous injection of vegetative cells into a mouse allows one to bypass the germination step required for spores, I don’t completely follow that one can conclude that PTS-Glu is required as a sensor for toxin production vs required for bacterial survival and replication in the host as a result of its role in carbon metabolism. Are the authors concluding that toxin production is key based on the timing of mice succumbing to infection by vegetative cells? Would vegetative cells that constitutively produce toxin bypass the need for PtsG?

**Part III – Minor Issues: Editorial and Data Presentation Modifications**

Reviewer #1: 1. A key finding in this study was a physical interaction between AtxA and PtsG (PTS-glu) identified using a co-purification protocol. Yet, the authors find quite few other proteins that also come down with AtxA-His other than PtsG. One issue that should be simple to address is that it was quite difficult to figure out what the proteins were identified in the supplemental table. It was hard to figure out from the ORF/Protein designations given. I’m guessing most have gene names that ould be used or presented. The total number of interacting proteins was a bit surprising, but the authors suggest it’s normal for a transcription factor. Is there support for that in the literature and, if so, they should reference here. However, it does place more emphasis on the validation protocol for the interaction (FRET/FLIM). I know even this amount of experimentation was quite a bit of work but was surprised that the authors didn’t try do a reciprocal pull down/co-purification with PtsG-His which would have been typical. Was it tried and failed? Are there other PTS components beyond PtsG that might also interact?

2. The authors address in the last paragraph of the Discussion their conclusions and model here in relation to those from the Koehler and Perego labs related to phosphorylation, particularly the Bier et al 2020 study. This paragraph should be expanded a bit to encompass the model of PTS impacting atxA expression through an unknown PTS regulator. Is there a role for transcriptional regulation of atxA in the model presented by the authors or in interpretation of the data? This discussion in important to try and bring together the full picture of AtxA and toxin regulation from the literature.

Reviewer #2: I only have minor comments and correction.

L194 “As seen in Figure 1C, omission of either glucose or 5% CO2 from the growth medium results in the loss of toxin induction.” It’s not in Figure 1C but rather 1E

L196, for clarity it should be precised that there is glucose in the medium “Therefore, we compared the AtxA interactome in bacteria grown with glucose and with and without external 5% CO2”

Figure 3A and B: In the interactome study, does the authors identified protein involved in phosphorylation/ dephosphorylation of AtxA?

Figure 3C, is the scale of the different images the same? The FRET-FLIM experiment could have been performed in presence of CO2-bicarbonate, especially because CO2-bicarbonate might modify the conformation of AtxA and consequently its aggregation but also its interaction with proteins, and especially because the authors identified interaction between AtxA and PtsG in condition with 5% CO2 (figure3A). This is also true for the supplementary figure 4D, where the faster lifetime decay observed in AtxA-EIIB-C4 alone might not be observed in presence of CO2 bicarbonate.

Line 238 “Despite PtsG being a critical glucose transporter in B. anthracis” To confirm this, it could have been interesting to measure glucose uptake in the ptsG mutant.

Line 258 add figure 4B

Line 264 Please correct CRH motif by CIXR motif

Line 294, the author injected spores of the ΔptsG and ΔptsG/Δpyc strains, but did they checked the ability to germinate of this spore?

What is the diet of the mice used in this study? Does the diet of the mouse might influence the outcomes? High glucose diet should increase toxin synthesis and consequently virulence.

Figure 6 did not show interaction between AtxA and pyc, what is the authors hypothesis about this interaction?

Materials and methods

Line 484 please correct average AD600 by average OD600

Line 520 please insert the correct reference instead of DOI

Line 555 please replace update by uptake

Legend figure 2A Line 824 “growth conditions in in RM” remove in

Reviewer #3: Minor comments

1. Line 99: this section title is a bit awkward. Suggest changing it to ‘Anthrax toxin production is dependent on a central carbon metabolism gene’

2. Fig. 1C: I am not familiar with the True/False designation used for 1C. It would be helpful to include some description of what this panel is showing.

3. Line 420: incomplete sentence, “Also, how these extracellular bacteria adapted their metabolic pathways to survive under different environmental conditions.”

PLOS authors have the option to publish the peer review history of their article (what does this mean? ). If published, this will include your full peer review and any attached files.

**Do you want your identity to be public for this peer review?** For information about this choice, including consent withdrawal, please see our Privacy Policy .

Reviewer #1: No

Reviewer #2: No

Reviewer #3: No

**Figure resubmission:**
---

## [Editor Report · Decision Letter 1]

18 Nov 2025

Dear Dr. Bothra,

We are pleased to inform you that your manuscript 'Environmental Regulation of Toxin Production in Bacillus anthracis' has been provisionally accepted for publication in PLOS Pathogens.

Best regards,

Bruce A. McClane

Academic Editor

PLOS Pathogens

Alice Prince

Section Editor

PLOS Pathogens

Sumita Bhaduri-McIntosh

Editor-in-Chief

PLOS Pathogens

orcid.org/0000-0003-2946-9497

Michael Malim

Editor-in-Chief

PLOS Pathogens

orcid.org/0000-0002-7699-2064
---

## [Editor Report · Acceptance letter]

Dear Dr. Bothra,

We are delighted to inform you that your manuscript, "Environmental Regulation of Toxin Production in Bacillus anthracis," has been formally accepted for publication in PLOS Pathogens.

Best regards,

Sumita Bhaduri-McIntosh

Editor-in-Chief

PLOS Pathogens

orcid.org/0000-0003-2946-9497

Michael Malim

Editor-in-Chief

PLOS Pathogens

orcid.org/0000-0002-7699-2064